# Design and Enhancement of a Fog-Enabled Air Quality Monitoring and Prediction System: An Optimized Lightweight Deep Learning Model for a Smart Fog Environmental Gateway

**DOI:** 10.3390/s24155069

**Published:** 2024-08-05

**Authors:** Divya Bharathi Pazhanivel, Anantha Narayanan Velu, Bagavathi Sivakumar Palaniappan

**Affiliations:** Department of Computer Science and Engineering, Amrita School of Computing, Amrita Vishwa Vidyapeetham, Coimbatore 641112, India; cb.en.d.cse17013@cb.students.amrita.edu (D.B.P.); pbsk@cb.amrita.edu (B.S.P.)

**Keywords:** air quality monitoring, air quality forecasting, fog computing, internet of things, smart fog environmental gateway, fog intelligence, optimized lightweight model, fog–cloud collaboration

## Abstract

Effective air quality monitoring and forecasting are essential for safeguarding public health, protecting the environment, and promoting sustainable development in smart cities. Conventional systems are cloud-based, incur high costs, lack accurate Deep Learning (DL)models for multi-step forecasting, and fail to optimize DL models for fog nodes. To address these challenges, this paper proposes a Fog-enabled Air Quality Monitoring and Prediction (FAQMP) system by integrating the Internet of Things (IoT), Fog Computing (FC), Low-Power Wide-Area Networks (LPWANs), and Deep Learning (DL) for improved accuracy and efficiency in monitoring and forecasting air quality levels. The three-layered FAQMP system includes a low-cost Air Quality Monitoring (AQM) node transmitting data via LoRa to the Fog Computing layer and then the cloud layer for complex processing. The Smart Fog Environmental Gateway (SFEG) in the FC layer introduces efficient Fog Intelligence by employing an optimized lightweight DL-based Sequence-to-Sequence (Seq2Seq) Gated Recurrent Unit (GRU) attention model, enabling real-time processing, accurate forecasting, and timely warnings of dangerous AQI levels while optimizing fog resource usage. Initially, the Seq2Seq GRU Attention model, validated for multi-step forecasting, outperformed the state-of-the-art DL methods with an average RMSE of 5.5576, MAE of 3.4975, MAPE of 19.1991%, R^2^ of 0.6926, and Theil’s U1 of 0.1325. This model is then made lightweight and optimized using post-training quantization (PTQ), specifically dynamic range quantization, which reduced the model size to less than a quarter of the original, improved execution time by 81.53% while maintaining forecast accuracy. This optimization enables efficient deployment on resource-constrained fog nodes like SFEG by balancing performance and computational efficiency, thereby enhancing the effectiveness of the FAQMP system through efficient Fog Intelligence. The FAQMP system, supported by the EnviroWeb application, provides real-time AQI updates, forecasts, and alerts, aiding the government in proactively addressing pollution concerns, maintaining air quality standards, and fostering a healthier and more sustainable environment.

## 1. Introduction

The escalating issue of air pollution poses a significant concern due to its widespread impact on human health and the environment, eliciting attention from industrialists, governments, academicians, and communities worldwide. According to the World Health Organization (WHO), over 92% of cities worldwide do not meet the established air quality guidelines. Air pollution is a major problem, particularly in developing nations like India, which ranks third in greenhouse gas emissions after China and the United States [1]. Despite government efforts, air quality levels have significantly worsened over the years due to various factors, including industrialization, urbanization, weather conditions, geographical features, and vehicular emissions. Air pollution is considered the most significant environmental health threat, where 9 out of 10 people breathe polluted air, causing seven million deaths globally every year [2,3]. It increases the risk of chronic respiratory diseases, impairs cognitive function, contributes to cardiovascular diseases and cancer, and increases susceptibility to viral infections like COVID-19 [4]. Elevated pollution levels seriously affect public health, the climate, the economy, and the ecosystem [5]. With 68% of India’s population projected to live in urban areas by 2050, the current monitoring infrastructure is insufficient. The Central Pollution Control Board (CPCB) has announced that India plans to double the air quality monitoring stations to address these existing challenges. The comprehensive monitoring requirements and the pressing pollution threats necessitate the development of an efficient IoT-based architecture for air quality monitoring and forecasting systems using state-of-the-art technologies. This would enable policymakers to create strategies for pollution prevention and preemptive actions, thereby improving public health, environmental protection, urban planning, public awareness, economic benefits, and regulatory support in smart cities.

Experts estimate that 75 billion Internet of Things (IoT) devices will be connected to cyberspace by 2025 [6]. The advancements in the IoT and Artificial Intelligence (AI) have fueled the development of smart city applications, generating vast and diverse datasets. Efficiently analyzing these growing data in real time and making timely decisions is essential for improving urban life. While Cloud Computing offers substantial computational and storage capabilities for the data, it is limited by computational overhead, bandwidth constraints, and latency [7,8]. To overcome these limitations, Fog Computing (FC) was introduced. FC extends Cloud Computing by bringing computation, communication, storage, and networking closer to IoT data sources at the network’s edge in a decentralized manner [9]. By enabling data processing at fog gateways or nodes positioned between the edge (terminal) and the cloud layer, rather than sending all data directly to the cloud [10], FC reduces reliance on the cloud connections and minimizes potential data flow interruptions [8]. FC offers benefits like low latency, minimized bandwidth usage, real-time decision making and responses, contextual awareness, and scalability, effectively satisfying the Quality of Service(QoS) requirements for smart city applications [11].

To address the challenges in real-time applications, researchers have turned to advanced technologies such as IoT and Fog Computing. Despite the growing significance of FC, the majority of the existing Air Quality Monitoring (AQM) solutions are cloud-based, leading to high monitoring and communication costs. These solutions often lack real-time and accurate multi-step forecasting and timely decision making and fail to optimize Deep Learning (DL) models for the efficient deployment on fog nodes. To address these challenges, this paper proposes a novel Fog-enabled Air Quality Monitoring and Prediction (FAQMP) system, featuring a low-cost AQM node with efficient communication and a Smart Fog Environmental Gateway (SFEG) that incorporates efficient Fog Intelligence to enhance real-time decision support in smart cities. The system effectively integrates state-of-the-art technologies, including Fog Computing (FC), IoT, LPWAN, and Deep Learning (DL). Specifically, Fog Intelligence enables the execution of DL models on fog nodes at the network’s edge. However, the resource constraints of fog nodes pose challenges, leading to innovations in model optimization, energy efficiency, and real-time data processing [12]. With this concern, this paper highlights achieving efficient Fog Intelligence by optimizing DL models for fog nodes like SFEG through an optimized lightweight DL-based Sequence-to-Sequence (Seq2Seq) Gated Recurrent Unit (GRU) model. This model strikes a balance between model performance and computational efficiency, thereby enabling efficient utilization of SFEG resources.

The proposed FAQMP system features a hierarchical, three-layered architecture with the sensing layer, the FC layer, and the cloud (CC) layer. At the sensing layer, an AQM node is equipped with a customized PCB and low-cost sensors to acquire the major air pollutants data that contribute to Air Quality Index (AQI) levels, including particulate matter 2.5 (PM_2.5_), particulate matter 10 (PM_10_), nitrogen dioxide (NO_2_), sulfur dioxide (SO_2_), carbon monoxide (CO), ozone (O_3_), and meteorological data, including temperature, pressure, humidity, wind speed (WS), wind direction (WD), and solar radiation (SR). This data is transmitted to the FC layer via LoRa, a Low-Power Wide-Area Network (LPWAN) technology known for its long-range capabilities, cost-effectiveness, and low power consumption [13,14]. The FC layer utilizes Raspberry Pi 3 Model B+ [15] as the Smart Fog Environmental Gateway (SFEG) embedded with Fog Intelligence. The SFEG processes and manages data, provides real-time forecasting, and supports decision making. It includes an Early Warning System (EWS) that detects anomalies and triggers alerts, enabling preemptive actions and rapid responses. The processed data are then sent to the cloud for long-term storage and complex analysis using MQTT. Additionally, the EnviroWeb application provides users with real-time air quality data, forecasts, trends, and alerts, helping them make informed decisions and improve public health and safety.

The proposed system addresses the challenges of DL model deployment for real-time air quality forecasting through fog–cloud collaboration. This collaboration allows to train and optimize the DL model in the cloud and then transfer it to the SFEG to facilitate Fog Intelligence. Initially, a DL-based Seq2Seq GRU Attention model is proposed, which outperforms the state-of-the-art DL baseline models by effectively capturing temporal dependencies in multi-step air quality forecasting. This initial DL model is further optimized through post-training quantization to be lightweight and efficient for deployment on resource-limited fog nodes like SFEG. This optimization enables efficient Fog Intelligence by reducing model size and computational latency while maintaining forecast accuracy.

To the best of our knowledge, no existing studies have developed an end-to-end air quality monitoring and forecasting system by incorporating efficient Fog Intelligence. The proposed FAQMP system effectively integrates Fog Computing with LPWAN, creating a decentralized architecture that reduces monitoring and communication costs. Specifically, it introduces an efficient Fog Intelligence on the SFEG using an optimized lightweight DL model, enabling real-time processing, accurate multi-step forecasting, and timely early warnings and event response to anomalous events, which are crucial for decision support in smart city environments to improve public health and safety.

The main contributions of the work are summarized below:Proposed a novel Fog-enabled Air Quality Monitoring and Prediction (FAQMP) system leveraging IoT with Fog Computing, LPWAN, and DL, aiming to support real-time and low-cost monitoring with accurate forecasting for decision support in smart cities.Developed a Smart Fog Environmental Gateway (SFEG) that introduces efficient Fog Intelligence in the FC layer through fog–cloud collaboration.Developed a user-friendly web application, namely EnviroWeb, to present real-time air quality AQI trends, forecasts, early warnings, and alerts to the users.Proposed a DL-based Seq2Seq GRU Attention model for multivariate multi-step time series air quality forecasting. The model demonstrates superior performance and stability in forecasting air quality for multiple time steps in comparison against baseline models.Developed an optimized lightweight DL model that facilitates efficient Fog Intelligence on the SFEG by striking a balance between computational efficiency and model performance.

The remainder of the paper is structured as follows: Section 2 provides an overview of the related works of air quality monitoring and forecasting systems and discusses the technologies enabling Fog Intelligence in IoT environments. Section 3 presents the proposed framework; the tools and technologies involved; the implementation of fog–cloud collaboration; the scalability aspects; and the real-world impacts. Section 4 describes the methodology for multivariate multi-step air quality forecasting. Section 5 presents the experimental evaluation and results of the proposed optimized lightweight DL model tailored for efficient Fog Intelligence. Finally, Section 6 discusses the concluding remarks and outlines future work.

## 2. Related Works

This section reviews related works on IoT-based air quality monitoring and forecasting systems and delves into methods and technologies that facilitate efficient Fog Intelligence in IoT environments. Each of the following subsections discusses the challenges, research gaps, and contributions to the proposed approach.

### 2.1. Air Quality Monitoring (AQM) Systems

The need for efficient AQM arises from the impact of air pollution on public health and the environment. Government agencies have established a quantitative tool called the Air Quality Index (AQI) [16,17] to express air quality levels and inform the public about the degree of pollution. Each country has its own standards to define the AQI. For example, in India, the AQI is calculated from six major pollutants: PM_2.5_, PM_10_, CO, NO_2_, SO_2_, and O_3_. The AQI converts the pollutant values into a single index value in the range 0–500 [14] classified under one of the six categories: good, moderate, sensitive, unhealthy, very unhealthy, and dangerous, as presented in Table 1. The higher the AQI levels, the greater the severity of pollution and its impact on human health.

The individual AQI for the pollutants is denoted by *I*_1_, *I*_2_…, and *I_n_* and calculated using the linear interpolation method as in Equation (1).
(1)$$I_{n}=\frac{\left(I_{high}-I_{low}\right)}{\left(C_{high}-C_{low}\right)}\times \left(C-I_{low}\right)+I_{low}$$

$C_{high}$—Breakpoint concentration greater than or equal to the given concentration.

$C_{low}—$Breakpoint concentration less than or equal to the given concentration.

$I_{high}—$AQI value corresponding to $C_{high}$.

$I_{low}—$AQI value corresponding to $C_{low}$.

C—Concentration of the pollutant.

*n*—Total number of pollutant.

Numerous studies have advanced IoT-based air quality monitoring (AQM) systems using a range of sensors, microcontrollers, and communication technologies. For instance, Dhingra et al. [18] developed the IoT Mobair kit with gas sensors, Arduino Uno, ESP8266 Wi-Fi module to provide real-time air quality data, and a map with pollution levels using Ubidots services. Similarly, Laskar et al. [19] created a framework that displays air quality data from sensors installed on roadways, recommending routes with the lowest AQI through an Android application for health-conscious users to make informed decisions. Alam et al. [20] introduced a low-cost system for measuring various environmental parameters but did not monitor all major pollutants responsible for AQI calculations or offer real-time alerts. Kelechi et al. [21] emphasized the affordability and accessibility of their low-cost AQM system for widespread deployment. While Kumar et al. [22] developed a low-cost, decentralized, and portable AQM system called AIRO that monitors air quality values, sends data to the AWS cloud, and forecasts AQI, it overlooks the meteorological parameters that likely influence air quality levels. Few studies have considered monitoring the meteorological parameters; for instance, Bobulski et al. [23] designed a monitoring system, highlighting the relationship between atmospheric pressure and particulate matter (PM10 and PM2.5). A recent work by Cabrera et al. [24] developed an affordable AQM system that monitored both meteorological parameters and pollutants. They stress that meteorological factors such as temperature, pressure, humidity, wind speed, and wind direction significantly influence the dilution and diffusion of air pollutants, subsequently affecting their distribution and concentration levels across seasons. Monitoring meteorological parameters in AQM is crucial for understanding the complex interactions between atmospheric conditions and air quality, as well as for grasping how seasonal changes impact air quality, which enables more accurate predictions and air pollution management. Similarly, Asha et al. [25] introduced the ETAPM-AIT model, which considered monitoring meteorological parameters and pollutants to provide real-time updates and alerts on hazardous conditions. Arroyo et al. [26] designed a portable system that tracks air quality, but it suffers from a limited communication range due to its reliance on short-range communication technologies, reducing its effectiveness for wide-area monitoring. A similar work by Barthwal et al. [27] used Bluetooth in the developed IoT sensing system to transmit air quality data to an Android app that limits its coverage and effectiveness.

While the aforementioned systems are cloud-based, they lack real-time processing capabilities at the edge of the network, which is crucial for an immediate response to air quality changes. Furthermore, transmitting data directly to the cloud becomes expensive due to the associated costs of data transmission, processing, and storage. In addition, most of the AQM systems employ short-range communication technologies like Bluetooth, Zigbee, and Wi-Fi, leading to limited coverage range, higher power consumption, and increased susceptibility to interference in urban environments. Regarding the hardware platforms in AQM systems, many utilize RPi [28,29,30] as the end device, which is expensive compared to microcontrollers like NodeMCU, Arduino Mega, and Arduino Uno, especially when deploying multiple nodes in the same network. IoT advancements have resulted in the development of low-cost sensors that are compact, lightweight, and energy-efficient. These improvements have facilitated the comprehensive deployment of AQM systems. Despite the popularity of low-cost sensors, there are reliability issues due to environmental sensitivity, instability, and manufacturing defects [31]. To address these challenges, sensors need to be calibrated to ensure accurate air quality measurements. For example, Koziel et al. [32] developed a simplified field calibration method for low-cost particulate matter (PM) sensors. The growing sensor market has driven advancements in calibration techniques to manage sensor drifts under harsh conditions and help improve the accuracy and reliability of sensor data, which are crucial aspects for effective AQM and analysis.

To overcome the limitations of cloud-based AQM systems discussed above, Fog Computing-based AQM solutions [33,34] with LPWAN [35] are gaining significant emphasis. FC brings storage, computation, and networking closer to the edge for faster processing, reduced latency, reliability, resilience to network failures, and minimized bandwidth consumption and reduced reliance on the cloud. For instance, Senthil Kumar et al. [36] proposed a fog-based intelligent air pollution monitoring system and analyzed the influence of meteorological parameters on AQI trends but did not integrate LPWAN for efficient communication with the fog node. In contrast, Santos et al. [37] emphasized the significance of integrating LPWAN in FC. Their fog-based system with anomaly detection, deployed on a Bpost delivery car in Antwerp, monitored PM_1_, PM_2.5_, and PM_10_ and promptly alerted citizens to elevated pollution levels. The study found that integrating FC improved response times compared to cloud-only solutions and underscored the importance of integrating LPWAN technologies in FC for efficient, real-time, and scalable AQM in smart cities.

LPWAN technologies overcome the limitations of short-range communication by supporting long-range transmission, low-power consumption, decentralization, prolonged network battery life, and energy efficiency, making them well suited for the resource-constrained fog environments. LoRa utilizes chirp spread spectrum modulation and facilitates robust communication over long distances while maintaining resistance to interference and noise, making it a suitable and cost-effective communication technology for AQM. Among LPWAN technologies, LoRa emerges as a promising solution [38] compared to Narrowband IoT (NB-IoT) [39] and Sigfox for smart city applications like AQM [40].

Despite significant advancements in AQM, several research gaps require attention. As summarized in Table 2, most existing AQM systems are cloud-based systems that capture only a few pollutant parameters that contribute to AQI, lack meteorological data, use short-range communication technologies, fail to adopt gateways as the fog node for intelligent processing at the edge, and ignore real-time air quality forecasting with timely early warnings for decision making in smart cities. Furthermore, these systems rarely integrate LPWAN and Fog Computing (FC) to meet real-time requirements. To address these gaps, our work effectively integrates Fog Computing, LPWAN, and Cloud Computing technologies to develop a low-cost and efficient FAQMP system. It addresses the limitations of short-range communication technologies by using LoRa, effectively captures the pollutants along with the meteorological parameters, and introduces a Smart Fog Environmental Gateway that introduces intelligence for real-time forecasting and early warnings with event response to varying environmental situations. By leveraging the capabilities of Fog Computing and LoRa, the system will benefit from improved performance, real-time decision making, low latency, reduced costs, conservation of bandwidth, contextual awareness, and reliability. Additionally, the real-time EnviroWeb application presents live air quality data, AQI levels, historical trends, forecasts, and alerts to the stakeholders. Moreover, the proposed AQM system will serve as a foundation for accurate air quality forecasting.

### 2.2. Air Quality Forecasting Systems

The impact of air pollution on public health and the environment is substantial. Therefore, accurate multi-step air quality forecasting is crucial for decision making, early warnings, and strategies to address pollution concerns and maintain safe AQI levels in smart cities. Air quality forecasting involves estimating pollutant concentrations for future time steps based on historical air quality data. However, accurate multi-step forecasting is challenging due to the influence of environmental, meteorological, and various other conditions [41] on pollutant patterns. The common approaches for air quality forecasting include statistical and data-driven methods, such as Machine Learning (ML) and Deep Learning. Although traditional statistical approaches like Multiple Linear Regression (MLR), Moving Average (MA), Autoregressive (AR), Autoregressive Moving Average (ARMA), and Autoregressive Integrated Moving Average (ARIMA) [42,43] are suitable for short-term air quality forecasting, their intrinsic linearity assumptions limit their ability to solve nonlinear problems, leading to high errors and poor model stability. Meanwhile, AI gained importance due to its capability to handle non-linearity in time series data. The studies in [44,45,46] utilized AI-based ML techniques like Support Vector Regression (SVR), Gradient Boost Regressor (GBR), Adaboost, and Stacking Ensemble to enhance forecasting accuracy. Although ML approaches have shown promising results, they fail to capture complex time series patterns and long-term dependencies in time series data affecting the forecast accuracy.

Later, DL methods emerged as a promising solution to model and forecast time series data in academia and industry. With the ability to perform non-linear mapping, self-adapt, and model complex relationships between inputs and output variables, DL enhances the capability to effectively capture temporal trends and achieve robust prediction performance [47]. The commonly used DL methods for air quality forecasting are Recurrent Neural Networks (RNNs), LSTM, and GRU [48,49]. While RNN is effective for sequential analysis, it faces a gradient explosion problem that is addressed by LSTM. LSTM addresses long-term dependency challenges in capturing time series data [50,51,52,53]. For example, Athira et al. [54] compared RNN models, finding that GRU slightly outperformed RNN and LSTM in predicting PM_10_. Similarly, Lin et al. [55] explored five GRU-based predictive models, which exhibited superior performance over SVR, GBT, LSTM, and Aggregated LSTM (ALSTM) models. However, GRU-based models demonstrated good performance in forecasting air quality across various scenarios, including seasonal variations, spatial distributions, and temporal patterns. They introduced an ensemble model called Multiple Linear Regression-based GRU (MLEGRU). In addition, Liu et al. [56] mentioned that shallow and DL predictors have evolved into hybrid approaches for enhanced forecasting performance. Hybrid forecasting models are based on combinations of multiple DL models to solve time series air quality data that have variations in time and space. It combines the advantages of multiple individual models. Several studies [57,58,59] have demonstrated that a hybrid RNN forecasting model yields better performance due to its capability to deal with complex and non-linear datasets, while also adapting to various forecasting requirements.

Furthermore, research on meteorological conditions influencing air quality levels primarily considered the correlation between air quality and meteorological conditions, often overlooking their influence in forecasting. To address this gap, our previous work [14] analyzed the influence of meteorological factors in forecasting and illustrated how incorporating these parameters enhanced accuracy. In addition, recently, researchers [60,61] also assessed the influence of meteorological conditions in forecasting air quality levels.

Most of the air quality forecasting studies primarily focused on single-step air quality forecasting methods. However, these methods only forecast air quality levels for the immediate time step and fail to offer insights into future steps. Consequently, multi-step forecasting methods have been introduced to effectively capture the behavior of future hours air quality data and provide a more detailed and actionable view of future air quality conditions, enhancing both short-term responses and long-term strategies for pollution management. For example, Chang et al. [62] implemented an ALSTM model that outperformed GBTR, SVR, and LSTM to predict PM2.5 for the next 8 h. Du et al. [63] proposed a novel DAQFF model for multi-step forecasting of PM2.5 up to 6 h, utilizing one-dimensional CNNs and Bi-LSTM. Kow et al. [64] suggested a CNN-BPNN model for regional multi-step-ahead forecasting that produced accurate forecasts compared to individual BPNN, RF, and LSTM models. Janarathan et al. [65] introduced an SVR-LSTM method for AQI prediction in a metropolitan city, outperforming the RNN, LSTM, and EMD-CNN models. Al-Janabi et al. [66] proposed a real-time intelligent forecasting system utilizing LSTM and PSO to predict primary pollutants for the next 48 h based on the information input from multiple stations in real time. Mokhtari et al. [67] developed a multipoint DL model based on Conv-LSTM to predict dynamic air quality levels, focusing on sudden pollution events. Furthermore, Hu et al. [68] proposed a Conv1D-LSTM and conducted extensive experiments to analyze the hourly PM_2.5_ and PM_10_ forecasts with a single- and multi-step prediction for the next 6 h.

However, the DL methods discussed for multi-step forecasting have limitations in extracting the long-term sequential characteristics of air quality data and forecasted only a few primary pollutants responsible for AQI. Therefore, state-of-the-art models with the Seq2Seq network, built on the encoder–decoder structure, address the issue by effectively handling temporal correlations in input and output sequences, leading to significant enhancements in multi-step forecasting tasks. Also, there are only limited studies on modeling temporal data with the encoder–decoder architecture for multi-step air quality forecasing. The encoder–decoder architecture employs a deep neural network (such as RNN or LSTM) to encode the input data into a fixed-length vector and utilizes another deep neural network to decode this vector into the target output sequence, generating predictions. For instance, Feng et al. [69] introduced an enhanced autoencoder (EnAutoformer) to improve air quality forecasting performance. Zhang et al. [70] developed an encoder–decoder model using an enhanced LSTM, known as read-first LSTM (RLSTM) as the encoder and LSTM as the decoder.

Despite the widespread adoption of encoder–decoders in multi-step forecasting, a limitation arises due to the potential loss of temporal information as the length of the input sequence increases during compression into a context vector with fixed-length encoding. To tackle this challenge, attention mechanisms are integrated into encoder–decoder models [71,72], enabling the model to focus on specific segments of the input sequence using attention weights and thus mitigating the risk of information loss. This mechanism assigns relative importance to each hidden state in the encoder, allowing the decoder to select relevant information from the input sequence to generate the output. It plays a crucial role in effectively extracting essential features and capturing long-range dependencies in long time series data, resulting in enhanced performance for multi-step forecasting. For instance, Dairi et al. [47] proposed a Variational Autoencoder (VAE) with multiple directed attention that outperformed RNN model variants in forecasting CO, NO_2_, O_3_, and SO_2_. Chen et al. [73] proposed the STA-LSTM, which incorporates an extreme value attention mechanism to mitigate the influence of sudden changes in air quality prediction. Li et al. [74] proposed an attention-based CNN-LSTM model that integrates an additional attention mechanism layer to capture spatiotemporal characteristics in the input data, demonstrating improved performance over the standard CNN-LSTM architecture. Du et al. [75] proposed a novel temporal attention encoder–decoder model based on Bi-LSTM to adaptively learn long-term dependencies and hidden correlation features for multi-step forecasting. Jia et al. [76] implemented a Seq2Seq framework with an attention mechanism to predict O_3_ for the next 6 h.

However, there are limited works that effectively explore the encoder–decoder structure with attention mechanism in temporal modeling of air quality for multi-step forecasting. The works mainly fail to forecast all of the six primary pollutants contributing to the AQI, ignoring the influence of meteorological parameters. A notable research gap is that the existing studies have overlooked the implementation of multi-step forecasting on fog nodes to facilitate real-time decision support, reduced latency, and high accuracy. With these considerations, we propose a Seq2Seq GRU Attention model for accurate multi-step forecasting of the primary pollutants in determining the AQI for future time steps. Furthermore, the model is optimized using a model compression technique, as discussed in the next section for efficient deployment on the SFEG.

### 2.3. Technical Enablers for Fog Intelligence in IoT Environmenst: Model Compression Techniques and Hardware Exploration

In the forthcoming years, IoT-based smart city applications are expected to increasingly integrate AI services into their core operations, with the training/inference ratio of DNN models projected to increase from 1:1 to 1:5 [77]. Consequently, significant advancements are anticipated through the adoption of AI-powered intelligence in Edge [78] and Fog Computing. Maccantelli et al. [79] suggest that embedded AI has proven effective in various IoT applications and is well suited for the monitoring of air quality in smart cities.

The Fog Intelligence paradigm brings the convergence of AI and Fog Computing to facilitate the execution of DL models partially or entirely to fog nodes located at the network’s edge. This paradigm is particularly useful in the IoT (Internet of Things) environments like air quality monitoring where data are generated and need to be processed in real time for timely decisions, reducing latency and improving efficiency. Typically, DL model deployment in the IoT is categorized into three main groups: (1) DL training and inference on the cloud; (2) DL training and inference on the fog; and (3) DL training on the cloud and inference on the fog. Group 1, utilizing the centralized cloud for both training and inference, introduces challenges like high latency, elevated communication costs, network congestion, and privacy concerns. Group 2 faces challenges in DL training due to resource constraints on fog nodes and in terms of memory, processing, and power. However, Group 3 distributes tasks by training DL models in the cloud and deploying them in the fog [80].

To enable Group 3 deployment, the fog–cloud collaboration allows CC and FC layers to work together for model training, inference, and data sharing and to enable Fog Intelligence. Fog Intelligence reduces latency, improves scalability and bandwidth savings, and provides faster decision making for real-time IoT applications. Despite the benefits it offers, a significant challenge lies in achieving efficient Fog Intelligence through model compaction, creating optimized lightweight DL models suitable for resource-constrained fog nodes with limited hardware and memory. This requires optimizing model size, forecasting accuracy, and execution time of the DL models. Key enablers for achieving this include model compression and appropriate hardware selection.

#### 2.3.1. Model Compression

Efficient Fog Intelligence involves deploying an optimized lightweight DL model on fog nodes. To achieve this, model compression methods such as quantization, pruning, and knowledge distillation can effectively reduce the size and complexity of DL models, thereby decreasing the memory footprint and computational requirements and reducing execution time while preserving model performance.

Quantization is a conversion technique that reduces the number of bits used to represent weights and activations by employing a smaller data type. This involves converting numerical values from a higher precision format, like 32-bit floating-point, to a lower precision format of 8-bit integers to represent layer inputs, weights, or both. Quantization techniques are classified into quantization-aware training (QAT) and post-training quantization (PTQ). PTQ includes dynamic range quantization, integer-only quantization, integer with float fallback quantization, and float16 quantization as depicted in Figure 1 [81].

Quantization offers three main advantages: reduced memory usage, reduced complexity of mathematical operations leading to faster inference time, and improved energy efficiency [82]. Frameworks like Nvidia TensorRT, TensorFlow Lite, and OpenVino support quantization for the efficient execution on edge and fog devices. For instance, TensorFlow Lite is a lighter version of the TensorFlow framework that allows quantization to an 8-bit integer (i.e., −127 to +127) for resource-constrained devices, enhancing power efficiency and reducing memory usage.

Pruning [83] involves removing the least significant parameters, such as weights and biases, to reduce the complexity and memory footprint of a neural networks. On the other hand, knowledge distillation entails the transfer of knowledge from a larger, more complex model (the teacher) to a simpler, smaller model (the student) by training the student to mimic its output, minimizing computation cost and memory usage, making it applicable for resource-constrained devices to allow efficient inference.

In summary, the model compression methods [84] transfer the capabilities found in large ensembles into smaller, more concise predictive models to achieve efficient Fog Intelligence. Due to the iterative nature of training, most compression techniques are applied post-training as in PTQ. Advancements in Fog Intelligence necessitate research advancements into the methods and effectively using them for fog deployment.

#### 2.3.2. Exploring Hardware for Fog Intelligence

Efficient Fog Intelligence requires deploying optimized lightweight DL models on suitable fog hardware. Recent advancements have enabled DL training and inference on resource-constrained devices, pushing intelligence to edge and fog nodes. It includes new hardware design and software frameworks enabling Edge and Fog Intelligence. The hardware supporting a range of AI algorithms include CPUs (Central Processing Units), GPUs (Graphics Processing Units), DSPs (Digital Signal Processors), FPGAs (Field-Programmable Gate Arrays), and specialized AI accelerators optimized for low-latency, power efficiency, and computational needs of AI at the edge [85]. Devices like the Raspberry Pi and Nvidia TX2 are considered potential edge and fog devices [86]. Table 3 provides a summary of a few potential edge and fog devices that enable the deployment of DL models, along with their specifications, showing that fog nodes are heterogeneous, particularly in terms of hardware capabilities.

The hardware specifications in Table 3 will help to select devices for proof-of-concept work related to edge and fog computing use cases. There are several AI accelerators and AI processors designed for edge and fog environments. Examples include the NVIDIA Jetson Series, Google Coral Edge TPU, and Intel Movidius Neural Compute Stick [87].

In addition, open-source AI frameworks are quite successful in developing, training, validating, and evaluating DL models, leveraging the acceleration offered by edge and fog hardware. Popular DL frameworks like TFLite, Caffe, and PyTorch have revolutionized Edge and Fog Intelligence. They operate best on GPUs and other specialized hardware devices to accelerate model training by reducing memory and computation requirements and model complexity, without compromising the model’s accuracy. However, it is quite challenging to select an appropriate framework while developing fog computing systems due to the varying performances and heterogeneity of fog hardware. It is inferred that the combination of the AI accelerator’s hardware efficiency and the flexibility of open-source frameworks enables developers to leverage the strengths of both aspects for efficient deployment of DL models on fog devices.

From the discussion, it is inferred that the efficient deployment of DL models on fog devices requires model compression methods and suitable hardware choices, serving as key enablers for efficient Fog Intelligence. This emerging research area presents numerous challenges and opportunities. Developing an efficient Fog Intelligence system involves DL model compression techniques, appropriate fog device hardware, and collaboration between software and hardware components. Therefore, hardware-aware software-level optimization offers promising research directions [9].

Moreover, there is limited research on developing optimized lightweight DL-based air quality forecasting models specifically tailored for deployment on resource-constrained fog nodes that can effectively handle computational and memory constraints for efficient Fog Intelligence. Our work addresses this gap by developing an optimized lightweight DL model using model compression (PTQ technique) that is suitable for fog nodes. For the proof of concept, the Raspberry Pi 3 model B+ is chosen as the fog node and enabled as a Smart Fog Environmental Gateway (SFEG). The Raspberry Pi 3 Model B+, a configurable single-board computer (SBC), is chosen over other fog devices presented in Table 3 due to its cost-effectiveness, processing and networking capabilities, compactness, connectivity, and low power consumption. The proposed optimized lightweight DL effectively strikes a balance between high forecast accuracy, significant file size reduction, and improved execution time on SFEG deployment. Moreover, efficient Fog Intelligence introduced using this model enables real-time processing of air quality data at the edge, accurate multi-step forecasting, optimized utilization of fog resources, and decision support through timely early warnings and event responses in real time, ensuring optimal AQI levels in smart cities.

## 3. Proposed Approach: A Fog-Based IoT Architecture and Fog–Cloud Collaboration in the FAQMP System

This section discusses the Fog Computing architecture, the functionalities of the SFEG, and the implementation of fog–cloud collaboration.

### 3.1. Architecture of the FAQMP System and Hardware Implementation

The proposed FAQMP system adopts a three-layered architecture: the sensing layer at the bottom, the FC layer in between, and the CC layer at the top, as depicted in Figure 2. Each layer has distinct functionalities that collaborate to achieve an efficient end-to-end IoT system, managing data processing and analysis across different layers. The architecture is explained in detail as follows:**Sensing layer:** The sensing layer serves as the foundation for monitoring air quality levels. The primary component of this layer is the AQM sensor node, designed with a PCB configured to function as an end device, as shown in Figure 3a. The PCB modularly integrates an array of dedicated low-cost sensors that acquire pollutant and meteorological parameters, along with a LoRa communication module connected to the controller unit. The controller unit is an Arduino Mega 2560, which is a low-cost, low-power, and resource-constrained microcontroller. Moreover, low-cost sensors have gained importance in facilitating dense deployments, greater coverage, and portability over traditional static monitoring systems. The sensors, as discussed in Table 4, are selected based on their cost, precision, accuracy, range, ability to monitor gases, lifetime, and compatibility with the controller. In particular, the sensors, including SDS011, MICS4514, MQ131, MQ136, BME280, pyranometer, and MPXV7002DP, measure the values of PM_2.5_, PM_10_, NO_2_, SO_2_, CO, O_3_, temperature, pressure, humidity, SR, WS, and WD. Moreover, due to variations between sensors in production, it is recommended to calibrate before deployment [27,31] to ensure accuracy in the measured values. Thus, the sensors in our AQM system are calibrated before data acquisition. For instance, Algorithm 1 presents the steps to pre-calibrate sensors like MQ136, where the coefficients x and y are extrapolated based on the characteristic curve presented in the datasheet [88]. The pre-calibration ensures that the sensor provides accurate and reliable readings of the measured gas.
**Algorithm 1:** Pre-Calibration of the MQ1361: R0 calculation (R0—Sensor resistance in the pure air);2: Rg calculation (Rg—Sensor resistance in the presence of a specific gas);3: Analog read sensor pin;4: Collect various samples and determine the aggregate (S);5: R0=S/clean air factor;6: Extrapolate coefficients x and y from datasheet;7: Estimate ppm values, ppm =x×RgR0y.


The AQM sensor node is statically placed 5 m from the ground, referring to the CPCB guidelines. It periodically samples the air quality data every minute. After acquiring the heterogeneous air quality and meteorological data, they are unified and appended with the sensor ID and timestamp to form a payload. Each payload comprises twelve floating-point 32-bit numbers (PM_2.5_, PM_10_, NO_2,_ SO_2_, CO, O_3_, temperature, pressure, humidity, SR, WS, and WD) equal to 48 bytes and a 16-bit integer (authentication number) equal to 2 bytes. The total number of payload bytes to be transmitted from the sensor to the FC layer is 50 bytes, which equals 400 bits. This payload is periodically transmitted to the FC layer through LoRa. The LoRa communication module Dorji DRF1276DM [89] is a unique module embedded with a Semtech SX1276 LoRa chip operating on an ISM frequency band of 433 MHz that is integrated with the controller unit to transmit data to the FC layer. The process by which the end device sends data to the upper-level device in the FC layer is referred to as uplinking. On the other hand, downlinking refers to the process of transmitting data from the fog layer to end devices. The uplinking and downlinking refer to the direction of data transmission between end devices and the FC layer. To ensure reliable data transmission from the AQM node to the FC layer, the adopted radio parameters for the LoRa link are transmission power of +14 dBm, spreading factor (SF) of SF9, coding rate (CR) of 4/5, and bandwidth (BW) of 250 kHz. The transmission power of +14 dBm provides a good trade-off between range and power consumption, ensuring reliable data transmission to the FC layer without the need for excessive power, thereby conserving battery life. SF9 maintains a balance between range and data rate by effectively handling a 50-byte payload. A CR of 4/5 offers robust error correction to ensure data accuracy despite interference. A BW of 250 kHz supports a higher data rate and facilitates quicker and efficient transmission of the payload while maintaining a good range. The adopted LoRa configuration strikes a balance among power consumption, data rate, and range.

The designed AQM system is a low-cost, low-power, multi-sensing, configurable, easily installable, and accurate system with long-range wireless data transmission capabilities to monitor air quality levels. Figure 3a shows the hardware of the proposed AQM sensor node that costs approximately USD 120 and is considerably less expensive than the traditional AQM systems for large-scale monitoring deployments.

**Fog Computing (FC) Layer:** FC is vital for air quality monitoring and forecasting due to its ability to offer computation and storage closer to the data sources with the benefits of minimized response time, optimized bandwidth efficiency, reliability, and reduced burden on the cloud. FC addresses the challenges of data processing, analysis, and transmission in dynamic environmental conditions. For the proof of concept, the proposed system utilizes a cost-effective Raspberry Pi 3 Model B+ as the fog gateway or fog node, as shown in Figure 3b. The fog gateway receives air quality sensor data from the sensing layer using LoRa module Dorji DRF1276DM. The received data are filtered and processed for further analysis. Fog intelligence is introduced by deploying an optimized DL model for on-device inference, enabling efficient processing by eliminating the need to constantly communicate with the cloud for processing. Moreover, the Early Warning System (EWS) detects anomalies and initiates an event response upon detecting dangerous AQI levels. Moreover, the real-time services offered by RPi for air quality data storage, management, communication, data analysis, early warnings, fog intelligence, and fog–cloud collaboration enable it to be a Smart Fog Environmental Gateway (SFEG). The services of the SFEG to manage the data and resources are detailed in Section 3.2. Furthermore, the forecast results and the air quality data are sent to the cloud using MQTT for historical storage and analysis.**Cloud Computing (CC) Layer:** The cloud layer at the top of the hierarchy centralizes and manages the data obtained from the SFEG in the FC layer. It offers a robust infrastructure to store and process historical air quality data, manage fog nodes, train complex DL models, and serve end-user applications. We chose the AWS platform as it offers a comprehensive suite of secure services like AWS IoT Core, AWS Lambda, DynamoDB, S3, CloudWatch, and Sage Maker, making it an ideal choice for our system requirements. Furthermore, a web application, namely EnviroWeb, is designed to present stakeholders with real-time air quality data, pollutant trends, AQI levels, forecasts, and early warnings using the data stored in the cloud.

Table 5 presents a summary of the tools, technologies, and methods utilized in the three layers of the FAQMP system.

### 3.2. Implementation of Fog–Cloud Collaboration in the Proposed FAQMP System

A significant distinction between the proposed FAQMP system and the traditional system lies in the incorporation of fog–cloud collaboration. Deploying a DL model via a cloud-based solution encounters communication latency, whereas edge- and fog-based solutions face challenges due to resource constraints. To address their standalone limitations, the fog and cloud platforms collaborate actively for model training and forecasting through fog–cloud collaboration. Fog–cloud collaboration entails integration and cooperation between the FC and CC layers for joint execution of DL tasks. As discussed earlier in Section 2.3, the deployment of a DL model for inference is categorized into three types based on its execution at different layers. However, the proposed system employs model training in the cloud and inferencing in the fog layer, fostering Fog Intelligence via fog–cloud collaboration. This approach leverages the computational power and resources of the cloud alongside the real-time capabilities of FC to achieve fast and responsive inference. The realization of fog–cloud collaboration that enables Fog Intelligence is achieved through various functionalities of the SFEG and cloud, as illustrated in Figure 4 and detailed below.

Node Authentication: The SFEG’s node authentication module authenticates the AQM sensor nodes to join the SFEG network for continuous transmission of air quality data. Initially, this module sends an authentication number to the AQM node in the sensing layer via LoRa downlink. The AQM node integrates the received authentication number with the collected air quality, forming a payload for LoRa uplinking to the SFEG. Daemons on the SFEG listen for incoming messages, extract live data, including the authentication number, and verify them. If the number matches, the AQM sensor node is authenticated and can send data, ensuring secure and reliable transmission.Data Handler: The SFEG’s data handler filters and preprocesses the received air quality data. Missing values are imputed using linear interpolation, and the AQI is calculated and appended to the preprocessed data. Preprocessing tasks like data cleaning, filtering, aggregation, and formatting enhance the data before analysis. The final prepared data, including PM_2.5_, PM_10_, NO_2,_ SO_2_, CO, O_3_, temperature, pressure, humidity, WS, WD, and SR, along with the AQI, are stored in the SFEG database to ensure seamless data recovery. SFEG data storage allows the system to remain stable and provide backup even during network outages and intermittent connectivity.

The cloud publisher transmits the data to the cloud over Wi-Fi using MQTT. MQTT [5] is a lightweight publish/subscribe messaging protocol suitable for resource-constrained devices to support low-bandwidth, high-latency, and unreliable network environments. The SFEG publishes air quality data in JSON format under the topic “sfeg/air_quality_data” to the AWS IoT Core broker, which processes the messages and stores the data in DynamoDB via an AWS Lambda function. This setup enables seamless integration of the developed Enviro Web application with DynamoDB for accessing air quality data.

Cloud Model Orchestrator: The cloud model orchestrator manages the training, optimization, and storage of the DL model in the cloud for deployment on the SFEG. At first, AWS Sage Maker facilitates model training with historical air quality data by leveraging the powerful compute instances. Based on the comparative analysis of various DL models presented, the Seq2seq GRU Attention model has good multi-step forecasting performance as analyzed from the results in Section 5.1.6. To make the model lightweight and efficient for SFEG, dynamic range quantization based on PTQ in model compression is applied, reducing its size and execution time while maintaining model accuracy. This optimized lightweight Seq2seq GRU Attention model, chosen based on the experimental results from Section 5.2, is then stored in AWS S3.Model Manager: By embracing fog–cloud collaboration, the model manager on the SFEG uses the model downloader to download the model from the AWS S3 bucket using the boto3 library. This model is deployed on the SFEG for inference, introducing efficient Fog Intelligence. Figure 5 illustrates the deployment pipeline [90] for the optimized DL model on a low-resource fog device like SFEG. The local air quality data are fetched, normalized, and fed into the model to generate multi-step forecasts for the next 3 h (12 time steps) at 15 min intervals in real time, without network delays. These live and meaningful forecasts are presented through the EnviroWeb application dashboard to the stakeholders.

Furthermore, the error assessment component determines the cumulative forecasting error of the deployed model over time using Root Mean Square Error (RMSE). If the cumulative error exceeds a predefined threshold, the model update component requests retraining in the cloud with updated air quality samples. The model is retrained and optimized, stored in AWS S3, downloaded by the model downloader, and deployed for new forecasts. This process in the model manager establishes a feedback loop where model retraining allows for regular updates to model parameters ensuring adaptation to seasonal changes and long-term air quality trends, thereby maintaining accuracy over time.Early Warning System (EWS) Handler: The EWS Handler analyzes the live data and forecasted data to detect anomalies and provide insights into the potential issues with the air quality data, including seasonal deviations. If any anomaly exists in the live data based on the AQI threshold defined, it is referred to as a live data anomaly. On the other hand, if an anomaly exists in the forecasted values, it is referred to as a prediction anomaly. Similarly, if an anomaly exists in both live and forecasted data, it is referred to as a discrepancy anomaly. While processing, if an anomaly is detected, the event variable is updated with the anomaly type (e.g., live data anomaly), and then the sub-module of EWS, namely the event response module, is activated as presented in Figure 4.Event Response: Based on the nature of the event, the SFEG’s event response module makes timely decisions and initiates appropriate actions to address the anomaly. These actions include information streaming, actuator control, and sensor network configuration.(a)Information Streaming: Timely alerts or notifications are sent to users via channels such as email, SMS, and EnviroWeb dashboards. This is crucial for providing immediate information about any detected anomalies, particularly those related to dangerous air quality levels. For testing purposes, we simulated a fire event where the AQI levels increased in the live data. This simulated spike activated the event response module, triggering an immediate reaction. As a result, an email alert was generated and received, as shown in Figure 6, demonstrating how the system works in real time to notify stakeholders about hazardous air quality conditions.(b)Sensor Network Configuration: During a live data anomaly, commands are sent to the controller of the AQM sensor node via LoRa downlinking. These commands adjust the AQM sensor node’s sampling rate to capture more detailed event information.(c)Actuator Control: During a live data anomaly, commands are sent to activate alarms, adjust ventilation systems, activate air purifiers, and control pollutant emission sources to maintain a healthy environment. This is crucial during time-critical hazardous events, such as dangerous air quality levels, fires, and gas leakages, to enable timely response by promptly reducing the severity of dangerous situations.

The event response helps to avoid exposure to abnormal pollutant levels, mitigate air pollution-related health risks, and address environmental concerns effectively.

Fog manager: The fog manager is mainly responsible for managing all the modules of the SFEG. It includes tasks such as resource allocation, data processing, data transmission, communication management, and overall coordination of activities within the FC environment.

In summary, the implementation of fog–cloud collaboration allows efficient utilization of data, knowledge, and resources across the fog and cloud layers, resulting in proactive decision making and improved responsiveness to address the requirements of real-time IoT-based air quality application services.

### 3.3. EnviroWeb Application

EnviroWeb, a user-friendly web application, empowers users with real-time air quality data and forecasting insights. By leveraging state-of-the-art technologies for air quality monitoring and prediction, EnviroWeb sets a new standard to help individuals and communities make informed decisions to protect their health and well-being and take a proactive step toward improved air quality levels. The main features of the Enviro Web application are presented below:Real-time Air Quality Data: The application offers real-time measurements of air pollution and meteorological data, as well as the Air Quality Index (AQI), as shown in Figure 7.Historical Data Analysis and Visualization: The application features customizable visualizations in charts and graphs to illustrate the historical pollution and AQI trends over a user-specified period (day, week, or month).Air Quality Forecast and Recommendations: The application presents AQI predictions for the next 3 h at an interval of 15 min. Based on the live and forecasted air quality data, decisions are made and recommendations are presented to the users related to health, travel, lifestyle changes, behavioral adjustments, and actions to reduce pollution levels.

Smart Alerts: When an anomalous event or hotspot is detected based on the live and forecasted air quality levels, alerts are presented in the dashboard as a part of the EWS Handler’s actuation (information streaming), as discussed previously in Section 3.2.Maps: The interactive map uses color-coded markers based on the AQI level of a specific location, allowing the users to navigate and explore the surrounding AQM station locations and determine AQI hotspots.Data Export and Sharing: The data export and sharing feature allows users to export the monitoring and forecast data.

The front-end application module is implemented through the Angular framework, a prevalent and extensively employed JavaScript platform to build responsive web applications. The backend uses AWS DynamoDB APIs to ensure seamless interaction with DynamoDB, allowing for efficient data management and retrieval of air quality data.

### 3.4. City-Wide Air Quality Management with FAQMP: Achieving Scalability and Real-Time Insights

Scaling the FAQMP system involves addressing the complexities and demands of monitoring and forecasting air quality on a larger, city-wide scale. This includes deploying thousands of AQM sensor nodes across multiple city locations, which creates a comprehensive network that delivers detailed and representative air quality data. This expanded network results in more accurate and reliable insights. This section explores the challenges and solutions associated with scaling the FAQMP system and highlights its real-world impact.

#### 3.4.1. Challenges and Bottlenecks in Scaling the Proposed FAQMP System

The challenges and potential bottlenecks in scaling the FAQMP system to a city-wide scale is outlined below:Data Volume and Management: Increased data from a growing number of sensors nodes will demand robust processing and storage solutions at fog nodes, such as the SFEGs.Resource Management: As the number of AQM sensor node increases, managing computational resources efficiently becomes crucial, requiring careful resource allocation.Communication Efficiency: Deployment of thousands of sensors leads to higher network traffic. Therefore, ensuring efficient communication among the sensor node, SFEG, and the cloud layer is essential for seamless data exchange and low-latency processing.Resource Constraints: Fog nodes like SFEG typically have limited CPU, memory, and storage resources compared to cloud servers, a fact which constrains their ability to process large volumes of sensor data and run complex algorithms.Real-time Processing: Handling high volumes of air quality data and complex forecasting models while maintaining minimal latency and timely responses is challenging.Model adaption and Performance: DL models must adapt to varying air quality patterns and environmental conditions by continuous retraining, which can be resource-intensive.Maintenance and Management: Managing and calibrating numerous sensors to ensure accurate air quality data are complex tasks.Cost Management: Scaling involves higher costs for hardware, installation, maintenance, and ongoing operations.Data Security: Ensuring the security and privacy of sensitive information is crucial.

While scaling the proposed FAQMP system, addressing the discussed issues is crucial to ensure effectiveness and efficiency in monitoring and forecasting air quality levels.

#### 3.4.2. City-Wide Implementation of the FAQMP System—Addressing the Scalability Challenges

By examining the challenges in scaling the FAQMP system from the previous section, the following discussion presents strategies to address these complexities, ensuring the successful deployment and operation of thousands of sensors in a scaled environment.

Let us consider scaling the proposed FAQMP system as in Figure 8 by deploying AQM sensor nodes across various city locations like residential areas, city parks, and industrial zones with each collecting the air pollution and meteorological data. The decentralized, hierarchical Fog Computing architecture uses distributed fog nodes, i.e., SFEGs in the fog layer to manage and process data from the AQM nodes. Each AQM node connects to a local SFEG in the intermediary FC layer and transmits data via LoRa, supporting efficient communication over long distances. Moreover, setting up LoRa in a mesh network will support a dynamic allocation of resources and routing of tasks based on current network conditions, offering significant benefits, including enhanced network resilience, improved data routing, and increased coverage. Each SFEG aggregates, preprocesses, and filters the data collected from the connected AQM sensor nodes, minimizing the volume of data transmitted to the central cloud. Local storage at the SFEGs helps temporarily retain sensor data, reducing data loss. These strategies help mitigate network congestion, optimize bandwidth, and reduce cloud storage requirements.

To maintain efficiency and responsiveness, Fog Intelligence employs lightweight DL models optimized for SFEGs, providing real-time air quality forecasts with minimal response time and improved performance. This approach allows scaling without overburdening the limited resources of SFEG. Furthermore, to keep the system adaptive to changing conditions, the DL models are periodically retrained. Since doing this directly on the SFEGs would be too resource-intensive, the system uses the proposed fog–cloud collaboration strategy where model retraining occurs in the cloud and real-time inference on the SFEG. This collaboration manages resources effectively, minimizes latency, and ensures efficient model updates.

For instance, if a fire is detected at any location of the city, the anomaly detection in the Early Warning System module alerts users through the EnviroWeb application, enabling timely responses and actions to critical events. The sensor network configuration captures detailed event information, and automated controls manage alarms and extinguishers. This automation scales operational capabilities without manual intervention, ensuring effective emergency responses. In addition, automated calibration adjusts sensors for drift and environmental changes. Moreover, cost-efficient hardware and technologies as in the proposed system help to manage the scaling expenses.

The above discussion clarifies how the FAQMP system can address scalability challenges such as managing large data volumes, optimizing resource allocation, maintaining efficient communication, ensuring real-time processing, model adaption, operational efficiency, and cost management, enabling it to effectively and efficiently monitor and forecast air quality levels across extensive urban areas.

#### 3.4.3. The Role of the FAQMP System in Shaping Public Health Policies and Urban Development

The proposed FAQMP system tackles air quality challenges through a multi-faceted approach and is significantly impactful in shaping public health policies and urban development based on the following:Public Health Protection: The system provides real-time AQI monitoring and forecasts via EnviroWeb, offering health advisories to vulnerable populations to reduce pollutant exposure. The recommendations will allow citizens to make informed decisions about outdoor activities and travel for an enhanced quality of life.Timely Warnings and Responses: The EWS module of the FAQMP system detects anomalous pollution events and triggers alerts via email and EnviroWeb and captures detailed information through adaptive sensor sampling. Identifying pollution sources enables targeted interventions and regulations to decrease emissions.Proactive Measures: Multi-step forecasting helps predict future hour’s air quality trends, allowing urban planners to implement preemptive strategies in real time and manage pollution peaks.Dynamic Policy Adaptation: The FAQMP system enables the formulation and adjustment of policies related to environmental regulations, emission standards, industrial regulations, and urban design based on the real-time AQI data.Enhanced Urban Planning and Resource Allocation: The FAQMP system will help urban planners identify pollution hotspots based on AQI levels, enabling optimized resource allocation for pollution control and urban infrastructure improvements, such as green spaces and buffer zones.Traffic Management: The data from the FAQMP system will support optimizing traffic flow and congestion management strategies, leading to reduced vehicular emissions.Community Engagement and Sustainable Development: The system promotes transparency and public awareness through EnviroWeb, fostering community involvement and sustainable development.Economic Benefits: The FAQMP system reduces healthcare costs and environmental damage through efficient monitoring and targeted interventions.

In summary, this section detailed the system architecture, and the hardware and software implementation of the FAQMP system, discussed its scalability, and highlighted its impact in urban planning.

## 4. Methodology

In this research, a GRU-based Seq2Seq architecture with an attention mechanism is investigated for multivariate multi-step forecasting of air quality levels over future time steps.

### 4.1. Gated Recurrent Unit (GRU)

The Gated Recurrent Unit (GRU) [91] is an advanced variant of an RNN, designed to address vanishing and exploding gradient problems by means of its gating mechanism. It effectively learns the long-term dependencies in time series data through its simpler internal structure and shorter training time compared to LSTM. The GRU architecture as shown in Figure 9 has an update gate, which regulates the retention and integration of past and new information, and a reset gate, which regulates how much of the previous state is to be forgotten. This GRU design supports handling both short-term and long-term dependencies efficiently.

The equations of the update gate (*z_t_*) and the reset gate (*r_t_*) in the GRU are presented in the following Equations (2)–(6):(2)$$r_{t}=\sigma \left(W_{r}.\left[h_{t-1},x_{t}\right]+b_{r}\right)$$
(3)$$u_{t}=\sigma \left(W_{u}.\left[h_{t-1},x_{t}\right]+b_{u}\right)$$
(4)$$\tilde{h_{t}}=tanh\left(W_{\tilde{h_{t}}}.\left[r_{t}*h_{t-1},x_{t}\right]+b_{h}\right)$$
(5)$$h_{t}=\left(1-u_{t}\right)*\;h_{t-1}+u_{t}*\tilde{h_{t}}$$
(6)$$y_{t}=\sigma \left(W_{o}.h_{t}\right)$$

$u_{t}\;and\;r_{t}$ represent the update gate and reset gate, respectively; $\tilde{h_{t}}$ denotes the candidate hidden state; $h_{t}$ represents the hidden state, and tanh is the hyperbolic tangent function. The activation function σ is used for both the forget and update gates.

*W_r_*, *W_u_*, and $W_{\tilde{h_{t}}}$ are the learned weight matrices associated with the reset gate, update gate, and candidate output, respectively. *b_r_*, *b_u_*, and *b_h_* are the bias vectors, and ∗ represents the element-wise multiplication.

### 4.2. Sequence-to-Sequence (Seq2Seq) GRU Attention Model

A Seq2Seq model is an encoder–decoder structure designed to map an input sequence to a target sequence. However, a traditional encoder–decoder model has challenges with temporal information loss, especially with longer sequences, which can degrade performance. To overcome these limitations, the proposed Seq2Seq GRU Attention model incorporates an attention mechanism. This attention mechanism selectively emphasizes relevant parts of the input sequence, allowing the model [92] to focus on crucial information and manage dependencies across long sequences effectively. The temporal attention layer is positioned as the interface between the encoder and the decoder. The architecture of the Seq2Seq model with attention mechanism is shown in Figure 10.

Let us consider the Seq2Seq GRU Attention model as a scholar working on a complex research article with the help of a comprehensive textbook. The encoder is analogous to the scholar, who reads and summarizes the textbook into a concise study guide, capturing the essential information and key points. The attention mechanism is like a highlighter that the scholar employs to identify the most critical sections of the study guide, emphasizing areas that are essential for comprehending the subject matter. Finally, the decoder is like the scholar writing the final article, utilizing the highlighted study material to produce a well-informed research paper. In air quality forecasting, the input data can be compared to the comprehensive textbook: the encoder uses GRU units to convert the input air quality data sequence into a compact, high-level summary known as the context vector. This summary captures essential patterns and trends from the historical data. The attention mechanism dynamically assigns different weights to various parts of the input sequence based on their significance in forecasting air quality levels. It highlights the most relevant parts of this vector for each forecasting step. The decoder employs GRU units to generate multi-step future predictions based on this highlighted information; i.e., it leverages the weighted context from the attention mechanism to make accurate predictions, considering the most relevant historical data for each forecast step. This process enables the model to convert extensive historical air quality data into actionable forecasts, like how a scholar turns extensive study material into a well-organized and insightful research paper.

The components of the Seq2Seq GRU Attention model are discussed in detail belowEncoder: The encoder is a GRU that processes the given input sequence $X=[x_{0,}x_{1},\ldots .x_{T}]$ to generate a sequence of hidden states [$h_{1},\;h_{2},\;....h_{T}]$, where *T* is the length of the input sequence. At each encoding time step t, the hidden state $h_{t}$ is updated by using both the input vector $x_{t}$ and the previous hidden state $h_{t-1}$, as illustrated in Equation (7).
(7)$$h_{t}=GRU\_Encoder(h_{t-1},x_{t})$$Attention Mechanism: The attention mechanism in the Seq2seq GRU-based Attention model allows focus on significant parts of the input sequence while generating output in the decoder, guided by attention scores.
Attention Score Calculation: The “attention score” or “alignment score” for each encoder hidden state *h_i_* is calculated using a scoring function. The attention score *e_t_*_,*i*_ indicates how much importance the decoder’s previous state places on the specific encoder state *h_i_*. The attention score is calculated as illustrated in Equation (8).
(8)$$e_{t,i}=S_{t-1}^{T}.h_{i}$$
where *e_t_*_,*i*_ is the attention score based on the dot product of the vectors that signifies the correlation between the previous decoder’s hidden state $S_{t-1}^{T}$ and i^th^ hidden state of the encoder *h_i_* at time step *t* and *T* refers to the transpose operation.
Attention weight calculation: Attention weights are the normalized version of the attention scores. After computing the attention scores $e_{t,i}$, a softmax function is applied to these scores to obtain the temporal attention weights $\alpha_{t,i}$ as displayed in Equation (9).
(9)$$\;\alpha_{t,i}=Softmax\left(e_{t,i}\right)=\frac{\exp(e_{t,i})}{\sum_{k=1}^{T}exp(e_{t,k})}$$
where *T* denotes the length of the input sequence.Context vector calculation: The context vector is a fixed-size representation of the input sequence, calculated by combining the encoder’s hidden states with the attention weights as illustrated in Equation (10):
(10)$$c_{t}=\sum_{i=1}^{T}\alpha_{t,i}.h_{i}$$

It represents a focused summary of the input sequence, with different elements weighted according to their relevance to the current decoding step *t*. Higher $\alpha_{t,i}$ values indicate an increased significance of the associated hidden state ($h_{i})$. This enables the attention mechanism to assign higher attention weights to elements in the input sequence that are more relevant to generating output at the current time step. The weighted summation mechanism in the context vector calculation allows the model to focus on different parts of the sequence dynamically and utilize relevant information from the hidden states of the encoder based on the attention weights during decoding.3.Decoder: The decoder is another GRU that reads the information from the context vector and its internal states to generate the output sequence. The context vector $c_{t}$ obtained from the attention mechanism is combined with the decoder’s previous hidden state $\left(s_{t-1}\right)$ and previous target output $\left(y_{t-1}\right)$, then fed to the GRU unit to compute the current hidden state $s_{t}$ as in Equation (11). $s_{t}$ acts as an initial point to compute the output sequence.
(11)$$s_{t}=f\left(s_{t-1},y_{t-1},c_{t}\right)\;\;\;\;$$
where *f* is the GRU function.
The output layer is a regression function that outputs the predicted value $y_{t}$. The decoder generates the output sequence at each time step *t* based on the current hidden state $s_{t}$, previous output $y_{t-1}$, and context vector $c_{t}$ as expressed in Equation (12).
(12)$$y_{t}=softmax\left({W.[c_{t},s}_{t},y_{t-1}\right]+b)$$*W* is the weight matrix; b is a bias vector, and ${[c_{t},s}_{t},y_{t-1}]$ represents the concatenation of the context vector, the current hidden state, and the previous output.

Similarly, the steps involved in the attention mechanism (step 2) and the decoder (step 3) are repeated until the maximum sequence length is reached.

## 5. Experimental Evaluation

We carried out experiments to create an optimized lightweight model for air quality forecasting, aiming to enable efficient deployment of Deep Learning (DL) models for Fog Intelligence on the Smart Fog Environmental Gateway (SFEG).

Experiment I involved determining an accurate multivariate multi-step DL-based air quality forecasting model by comparing the state-of-the-art methods. Subsequently, this determined model is made lightweight by conversion to a TFLite model and optimized by applying the PTQ technique based on model compression as discussed in Section 2.3.1. Experiment II focused on validating the performance of the optimized lightweight model. This involved analyzing the effects of different quantization methods on the initial model to ensure efficient deployment on fog nodes by reducing model size and lowering execution time while maintaining high model accuracy.

### 5.1. Experiment I: DL-Based Multivariate Multi-Step Forecasting

Multivariate multi-step air quality forecasting refers to a prediction modeling task that forecasts values of pollutant variables for future h time steps based on the multiple input variables (air quality and meteorological variables), where $h\in N^{*}$ denotes the forecasting horizon. If *h* = 1, the forecasting is simplified to single-step forecasting.

This section demonstrates the effectiveness of the proposed Seq2Seq GRU Attention model for the multivariate multi-step air quality forecasting model by comparing its performance with baselines using historical air quality data. Figure 11 shows the stages involved in carrying out Experiment I.

#### 5.1.1. Dataset Description

This study utilized historical air quality data obtained from the CPCB, a government-established AQM station located in SIDCO Kurichi, Coimbatore, Tamil Nadu, India (latitude: 22.544697, longitude: 88.342606) [93]. The dataset spans samples of one year, from June 2019 to June 2020, with 35,232 samples recorded at 15 min intervals, and is publicly accessible on the CPCB website. The investigative variables included the primary air pollutants and meteorological parameters such as PM_2.5_, PM_10_, NO_2,_ SO_2_, CO, O_3_, humidity, WS, WD, and SR. The dimensions of PM_2.5_, PM_10_, SO_2_, NO_2_, and O_3_ are expressed in micrograms per cubic meter (μg/m^3^), while CO is measured in milligrams per cubic meter (mg/m^3^). The EDA indicates that the pollutant levels exhibit an increase during the winter months of December, January, and February, in contrast to the monsoon months of June and July. This is because the meteorological conditions greatly influence the air quality levels and play a crucial role in enhancing the forecasting performance [14].

#### 5.1.2. Data Preprocessing

Data preprocessing is an important step in air quality modeling as it enhances the representation of collected data and improves model performance. It involves handling the missing values, addressing the outliers, normalizing the data, and performing a dataset split. The air quality data from AQM stations may reveal missing values and inconsistent measurements due to sensor malfunctioning, power failure, or influence of external and uncontrollable factors. In such cases, missing values increase the uncertainty of the data, making it difficult to effectively capture the temporal characteristics. As part of preprocessing, missing values are identified and imputed using the linear interpolation method as expressed in Equation (13), to ensure temporal continuity between the data points and improve the model’s effectiveness. We then identified the outliers using IQR and replaced them using linear interpolation.
(13)$$y^{*}=y_{1}+\frac{\left(t^{*}-t_{1}\right)}{\left(t_{2}-t_{1}\right)}\;\left(y_{2}-y_{1}\right)$$

*y** is the missing value at the time $t^{*}$, and ($t_{1},\;y_{1}$) and ($t_{2}$, $y_{2}$) are the data points of two known samples. The data $y_{1}$, $y_{2},\;\mathrm{and}\;y^{*}$ are in a straight line, and *t** is inside or outside the time interval [$t_{1},\;t_{2}$]. In addition, the data are normalized using min–max scaling in the boundary of [0, 1] via a linear transformation as expressed in Equation (14). The normalization eliminates the influence of measurement scale differences among different features in the dataset and improves the model convergence,
(14)$$x^{*}=\frac{x-x_{min}}{x_{\max}-x_{min}}$$
where *x** represents the normalized feature value and $x_{max}$ and $x_{min}$ represent the maximum and minimum values of the features, respectively.

The data are transformed into time series samples, and the train–test split is performed on the dataset, where the training set comprises 80% of the data and the testing set has 20% of the data. The training set is employed for model fitting, and the testing set is used to evaluate the performance of the model.

A multivariate time series of air quality data comprising variables of air pollutants and meteorological parameters recorded over consecutive time intervals is represented as $X_{i}=\left\{x_{i,j}\right\},\;\left(i=1,\;2,\ldots \ldots n;\;j=1,\;2,\ldots \ldots m\right)$, where *i* is the time dimension, *n* is the time series length, *j* is the feature dimension, and *m* represents the maximum value of variable dimension. $X_{i}$ vector refers to the values of air pollutants and meteorological variables at the *i*^th^ time step. The representation can be illustrated as a two-dimensional matrix, as shown in Equation (15):(15)$$X_{i}=\left[\begin{array}{cccc} x_{11} & x_{12} & \ldots & x_{1m}\\ x_{21} & x_{22} & \ldots & x_{2m}\\ \vdots & \vdots & \vdots & \vdots \\ x_{n1} & x_{n2} & \ldots & x_{nm} \end{array}\right]$$

During the training of a multivariate multi-step forecasting model, historical time series samples are utilized. Each sample consists of observations of all the features over a fixed window size, serving as the inputs. The output pair comprises samples with a sequence of target variables for the future time steps. In the multi-output strategy, a single function *F* maps the input to the output pairs, as in Equation (16) during training.
(16)$$\left[{\hat{X}}_{t+1},{\hat{X}}_{t+2},{\hat{X}}_{t+3},\ldots \ldots ,\;{\hat{X}}_{t+h\;}\right]=F\left(\left\{\;X_{t-w+1},\ldots \ldots ,\;X_{t}\right\}\right)\;\mathrm{where}\;t\in \left\{w,\ldots ,\;n-h\right\}$$
where *F* is a non-linear mapping function in the training phase, w is the sliding window size, ${\hat{X}}_{t+h\;}$ denotes the predicted value at time *t* + *h*, and $\hat{X}$ represents the target variable.

#### 5.1.3. Experimental Settings and Baselines

The experiments are conducted on an Apple MacBook Pro equipped with an M1 chip featuring an 8-core CPU and 8 GB of unified memory. The DL models are implemented in Python 3.7 using the Keras framework built on TensorFlow.

The state-of-the-art models, including GRU, Seq2Seq-GRU, GRU Autoencoder, GRU Attention, Seq2Seq LSTM Attention, Seq2Seq BiLSTM Attention, GRU-LSTM Autoencoder, LSTM-GRU, and LSTM-GRU Attention, are considered baselines for comparison against the proposed model.

#### 5.1.4. Evaluation Metrics

This section demonstrates the effectiveness of the proposed Seq2Seq GRU Attention model. To evaluate the predictive performance of the models investigated in the study, various statistical metrics, including Root Mean Square Error (*RMSE*), Mean Absolute Error (*MAE*), Mean Absolute Percentage Error (*MAPE*), Pearson’s Correlation Coefficient ($R^{2}$), and *Theil’s U*1 Index, are utilized, as depicted in Equations (17)–(21).
(17)$$RMSE\;\;\;\;\;\;=\;\sqrt{\frac{1}{N}\sum_{j=1}^{N}{\left(y_{j}-{\hat{y}}_{j}\;\right)}^{2}}$$
(18)$$MAE\;\;\;\;\;\;\;\;\;=\;\frac{1}{N}\sum_{j=1}^{N}\left|y_{j}-{\hat{y}}_{j}\right|$$
(19)$$R^{2}\;\;\;\;\;\;\;\;\;\;\;\;\;=\;1-\frac{\sum_{j=1}^{N}{\left(y_{j}-{\hat{y}}_{j}\;\right)}^{2}}{\sum_{j=1}^{N}{\left(y_{j}-{\bar{y}}_{j}\;\right)}^{2}}$$
(20)$$\;\;\;\;MAPE\;\;\;\;\;\;=\frac{1}{N}\sum_{j=1}^{N}\left|\frac{{\left(y_{j}-{\hat{y}}_{j}\;\right)}^{2}}{y_{j}}\right|\times 100\%$$
(21)$$Theil’s\;U1=\frac{\sqrt{\frac{1}{N}\sum_{j=1}^{N}{\left(y_{j}-{\hat{y}}_{j}\;\right)}^{2}}}{\sqrt{\frac{1}{N}\sum_{j=1}^{N}{\left(y_{j}\;\right)}^{2}}+\sqrt{\frac{1}{N}\sum_{j=1}^{N}{\left({\hat{y}}_{j}\;\right)}^{2}}}$$
where $y_{j}$ and ${\hat{y}}_{j}$ denote the actual and predicted values of *j*^th^ observation respectively, ${\bar{y}}_{i}$ denotes the average of the actual values, and *N* is the total number of samples. The models exhibit a better forecasting performance and fitting effect when the RMSE, MAE, MAPE, and *Theil’s U*1 values are smaller and *R*^2^ is closer to 1.

#### 5.1.5. Hyperparameter Tuning

Hyperparameter tuning is crucial for enhancing DL model performance by determining the optimal combination of parameters in the network architecture. It enhances model accuracy, optimizes processing time, and mitigates overfitting. There are several methods for hyperparameter optimization, like manual search, random search, grid search, Bayesian optimization, and metaheuristic algorithms. In our work, we adopted grid search with holdout validation for multi-step air quality forecasting. This approach evaluates the performance of the model used by systematically testing various hyperparameter combinations on the training set and verifying their effectiveness on the testing set. The hyperparameters of the proposed Seq2Seq GRU Attention model are the number of encoder and decoder units, activation function, optimizer, learning rate, forecast horizon, batch size, epochs, and window size. Table 6 displays the range of hyperparameter values and the optimal values obtained through a grid search for the proposed model. Finally, the proposed model is retrained using these optimal values on the entire training set to ensure accurate multi-step forecasting.

#### 5.1.6. Experimental Results: Analysis and Discussion

This section presents the experimental results analyzing the performance of the Seq2Seq GRU Attention model and baseline models in forecasting six primary pollutants responsible for the AQI: PM_2.5_, PM_10_, SO_2_, NO_2_, and O_3_ over multiple future time steps. During testing, we fed the model with pollutants and meteorological data as the input to forecast pollutant concentrations for the next three hours and outputted pollutants in twelve consecutive time steps (i.e., next three hours) at a 15 min time interval. All models are tested on the same test set, and the forecasting performance is assessed in terms of RMSE, MAE, R^2^, MAPE, IA, and Theil’s U1.

Although we analyzed the forecasting performance of all six pollutants, with page considerations, we present only the forecasting performance of the two significant pollutants, PM_2.5_ and PM_10_, in Table 7 and Table 8, at different stages of multi-step forecasting (1st time step to 12th time step). After analyzing the impact of forward forecasting step size on the forecasting performance for PM_2.5_ and PM_10_ in Table 7 and Table 8, and other pollutants, a common observation reveals that as the forecast time step increases, the RMSE, MAE, MAPE, and Theil’s U1 gradually increase, while R^2^ decreases for all the models. This indicates that multi-step prediction poses a significant challenge compared to single-step forecasting, primarily because the forecasting horizon significantly impacts accuracy owing to the uncertainty in time series air quality data arising due to weather patterns, environmental conditions, and other factors that affect air quality. Therefore, it is crucial to consider the forecasting horizon when evaluating the forecasting models to ensure their effectiveness in providing reliable predictions over different time steps. Nevertheless, the proposed model exhibits good performance in multi-step forecasting, even when the time step increases. The results of the proposed model are highlighted in bold.

From analyzing the forecasting performance results of PM_2.5_ in Table 7, the proposed Seq2Seq GRU Attention model achieved the best performance for long-term forecasting (t + 12), with the lowest RMSE (7.9083), MAE (5.4929), MAPE (27.4658), and U1 (0.177) and highest R^2^ (0.5309) against the baseline models. On the contrary, the GRU has the highest error rates for single-step-ahead forecasting (t + 1) and long-term forecasting (t + 12) when compared to the other models that are hybrid RNNs and encoder–decoder-based models. Similarly, analyzing the performance results of PM_10_ in Table 8, the proposed Seq2Seq GRU Attention model achieved the lowest RMSE (11.7805), MAE (8.3638), MAPE (29.323), and U1 (0.1925) and highest R^2^ (0.6212) compared to the baselines for long-term forecasting (t + 12). This is mainly because the attention mechanism in the Seq2Seq GRU architecture of the proposed system improves the long-term forecasting performance of the pollutants. In addition, the line graphs in Figure 12, Figure 13, Figure 14 and Figure 15 illustrate the performances of the various models compared in forecasting PM_2.5_ and PM_10_ concentrations over future time steps (t + 1 to t + 12). Moreover, it illustrates the ability of the proposed Seq2Seq GRU Attention model to maintain the lowest error and outperform the baselines, proving its effectiveness and stability in multi-step forecasting

In addition, Table 9 presents the performance metrics (RMSE, MAE, MAPE, R^2^, and U1) of various models for each pollutant (PM_2.5_, PM_10_, NO_2,_ SO_2_, CO, and O_3_), with values representing the average errors across twelve time steps (t + 1 to t + 12) for each model and pollutant. The last column displays the average performance across all pollutants, providing a comprehensive assessment of the model’s performance.

The results of Table 9 are discussed below:The traditional RNN model, GRU, exhibited the least average forecasting performance with an RMSE, MAE, MAPE, R^2^, and Theil’s U1 of 9.2129, 7.1219, 44.63, 0.081, and 0.2268, respectively. Moreover, the effectiveness of the GRU is improved through a hybrid RNN approach like the LSTM-GRU, which demonstrates better performance. Compared to the GRU, a hybrid LSTM-GRU has a better average forecasting performance across all the pollutants, where RMSE, MAE, MAPE, and Theil’s U1 are decreased by 10.89%, 11.32%, 16.54%, and 4.32%, respectively, while R^2^ is increased by 0.1052.Seq2Seq GRU exhibited improved average forecasting performance over the RNN variants (LSTM-GRU and GRU). This indicates that introducing an encoder–decoder into the RNN model is beneficial to enhance the forecasting performance. For instance, compared with the LSTM-GRU, the RMSE of Seq2Seq GRU decreases by 3.19%, the MAE decreases by 5.79%, MAPE decreases by 7.38%, Theil’s U1 decreases by 9.19%, and R^2^ increases by 0.1553.The forecasting performance of the Autoencoder model (GRU Autoencoder), a variant of the encoder–decoder is superior to the Seq2Seq GRU for all the pollutants. Despite this, the hybrid variant of AE (GRU-LSTM Autoencoder) has better performance than the GRU-AE. Compared with the GRU Autoencoder, the RMSE, MAE, MAPE, and Theil’s U1 of the GRU-LSTM Autoencoder decrease by 8.81%, 6.76%, 7.06%, and 7.24% respectively, and R^2^ increases by 0.0931.Moreover, adding an attention mechanism to the LSTM-GRU architecture, as seen in the LSTM-GRU Attention, led to an enhancement in forecasting performance. Compared with the LSTM-GRU, the RMSE, MAE, MAPE, and Theil’s U1 of the LSTM-GRU Attention model decrease by 12.76%, 15.45%, 19.23%, and 15.14%, respectively, and R^2^ is increased by 0.2022. The encoder–decoder-based attention variants (Seq2Seq LSTM Attention, Seq2Seq Bi-LSTM Attention, and Seq2Seq GRU Attention) exhibit improved performance over the Seq2Seq GRU. This indicates that introducing an attention mechanism overcomes the limitations of the traditional Seq2seq RNN models by dynamically focusing on relevant input sequences to capture contextual information critical for accurate predictions, mitigating information loss from fixed-length context vectors, addressing the vanishing gradient problem for effectively capturing long-range dependencies, and generating context-related forecasts with enhanced performance.The Seq2Seq Bi-LSTM Attention exhibits a similar average forecasting performance compared to the GRU-LSTM Autoencoder; the latter demonstrates better efficacy specifically for the pollutants PM_2.5_, NO_2_, and O_3_. However, the proposed Seq2Seq GRU Attention model demonstrates the best average performance across twelve time steps for each of the pollutants, as well as superior average forecasting performance across all the pollutants in comparison with the baselines. Compared with the Seq2Seq Bi-LSTM Attention, the average forecasting performance of the proposed model across all the pollutants in terms of RMSE decreases by 18.27%, MAE decreases by 33.83%, MAPE decreases by 33.51%, Theil’s U1 decreases by 18.95%, and R^2^ increases by 28.70%.The proposed Seq2Seq GRU Attention achieves the best average forecasting performance across six pollutants for 12 time steps, with an average RMSE of 5.5576, MAE of 3.4975, MAPE of 19.1991, R^2^ of 0.6926, and Theil’s U1 of 0.6926, as highlighted in bold in Table 10.

In addition, Figure 16 presents the bar chart illustrating the performance metrics (RMSE, MAE, MAPE, R^2^, and Theil’s U1) of various models across all pollutants over 12 time steps, showing that the Seq2seq GRU Attention model attains enhanced average forecasting performance over the baseline models.

To summarize the extensive experimental analysis, the results show that the Seq2seq GRU Attention model demonstrates superior forecasting performance by outperforming the baseline models for accurate multivariate multi-step air quality forecasting. The attention mechanism effectively captures the relationship between current and past time sequences to improve stability in long-term predictions of all the primary pollutants (PM_2.5_, PM_10_, NO_2,_ SO_2_, CO, and O_3_) responsible for determining AQI levels.

### 5.2. Experiment II: Evaluation of an Optimized Lightweight DL Model for Efficient Fog Intelligence

After determining an accurate AQ forecasting model (initial model) based on Experiment I, the study developed an optimized lightweight variant to facilitate efficient model deployment for Fog Intelligence on the resource-constrained fog node (SFEG). To achieve this, we converted the initial DL model (Seq2SeqGRUAttention model) built using TensorFlow (TF) into a TensorFlow Lite (TFLite) model. During this conversion, TFLite provided support for quantization techniques based on model compression. This section evaluates and compares the resulting TFLite models on the fog node, i.e., the SFEG, resulting from different post-training quantization techniques like dynamic range quantization, integer with float fallback quantization, full-integer-only quantization, and float16 quantization. This analysis helps to determine the suitable post-training technique resulting in an optimized lightweight Seq2Seq GRU Attention model that effectively reduces the model’s size and execution time while maintaining high forecast accuracy on the SFEG. This ensures that the model remains efficient and accurate.

The Raspberry Pi 3 Model B+, serving as the SFEG, is used to evaluate the performance of the TFLite models. It features a BCM2837 Quad-Core processor, 1 GB of RAM, built-in Wi-Fi, and Bluetooth 4.1 with BLE. Its support for TFLite, a lightweight version of TensorFlow, enabled the execution of DL models for real-time data processing in Fog Computing environments. TFLite is characterized by two main components:TFLite converter: The TFLite converter converts TF models into an optimized format by applying optimization techniques such as quantization, model pruning, and operator fusion to reduce model size and increase inference speed. It generates a TensorFlow Lite model file (.tflite) that contains the converted model in a format that the TFLite interpreter can handle.TFLite interpreter: The TFLite interpreter loads the TFLite model (optimized model), prepares it for execution, and enables on-device inferencing using the input data. It enables the efficient execution of TFLite models.

Together, the TFLite converter and interpreter enable an efficient deployment of TF-based DL models on fog devices with reduced memory footprint and inference.

We evaluated the resulting file size, execution time, and model accuracy of the TensorFlow Lite model with and without quantization. Table 11 compares the file size of the original TF model and the TFLite model that is not optimized on the SFEG. The original file size measures 1176 KB, while the TFLite version is significantly smaller at 397 KB, representing a reduction of three times in size. File size reduction is a key step for resource-constrained devices with minimal storage.

Furthermore, size reduction is achieved through post-training quantization. With the TFLite model’s size of 397 KB as a reference, dynamic range quantization achieves a reduction of 70%, full-integer quantization achieves about a 68% reduction, and float16 quantization results in around a 48% reduction, as shown in Figure 17. Based on these findings, dynamic range quantization outperforms other quantization techniques, albeit only slightly surpassing full-integer quantization in terms of file size reduction.

The execution time to forecast the test data is measured, and the results are presented in Table 12. The original TF model has the highest execution time of 323.5977 s, as there is no model compression involved. Among the TFLite models with and without quantization, the TFLite model without quantization has the lowest execution time, as there is no quantization involved. Moreover, when considering the quantized TFLite models, dynamic range quantization has a slightly higher execution compared to the TFLite model without quantization because of the additional processing required for quantization. However, dynamic range quantization offers a good balance between model size reduction and execution. Although full-integer quantization can achieve a 69% reduction in file size compared to the TFLite model that is not quantized, it has a higher execution time compared to other methods, with an execution time of 68.8838 s. In addition, float16 quantization has an effective execution time with a latency of 54.7483 as compared to the other TFLite models with quantization.

In addition to considering model size and execution time, we also evaluated the accuracy of the model for post-training quantization. Table 13 illustrates how quantization affects the average model accuracy of the six pollutants across 12 time steps for the original TF model (i.e., Seq2Seq GRU Attention model determined from Table 9) and TFLite models. If the focus is mainly on model accuracy, opting for original TF model or TFLite without quantization can be considered. However, it is not the most effective option to reduce model size and improve execution time for efficient execution on the fog nodes. Hence, TFLite models with post-training quantization are considered. The TFLite models offered a file size reduction compared to the original TF models, while showing varying levels of execution time and accuracy. Dynamic range and float16 quantization methods maintain model accuracy like the TFLite model without quantization. In addition, the full-integer quantization method has lower accuracy compared to other methods. Specifically, the dynamic range quantization outperforms other TFLite models in terms of model accuracy and file size reduction, but with a slightly longer execution time than float16 quantization.

On the other hand, float16 quantization strikes a balance between accuracy and execution time. While it may offer slightly lower accuracy compared to dynamic range quantization, it compensates by providing the advantage of the shortest execution time. Additionally, float16 quantization achieves only a moderate reduction in file size. Moreover, full-integer quantization offers a good file size reduction but impacts model accuracy and execution time; i.e., file size reduction comes at the expense of model accuracy and longer execution times impacting the real-time responsiveness of the SFEG.

Based on the experimental results, applying dynamic range quantization to the Seq2Seq GRU Attention results in an optimized lightweight model that maintains good model accuracy, offers a significant reduction in file size, and minimizes execution time, striking a good balance between performance and computational efficiency compared to other post-training quantization methods. The deployment of this optimized lightweight model on the SFEG will enable efficient Fog Intelligence and enhance the effectiveness of the FAQMP system through accurate multi-step air quality forecasting, timely early warnings through EnviroWeb and event response with reduced latency, and optimized fog resource utilization.

## 6. Conclusions and Future Works

Compared to the traditional air quality monitoring systems that process data in the centralized cloud, this study proposes a novel Fog-enabled Air Quality Monitoring and Prediction (FAQMP) system, based on IoT, Fog Computing, and Deep Learning for efficient real-time decision support in smart cities. Fog Computing brings storage, computation, and networking closer to the edge for faster processing, reduced latency, reliability, and minimized bandwidth consumption, reducing the burden of the cloud. In the three-layered architecture of the proposed FAQMP system, the sensing layer has a cost-effective AQM node designed with a customized PCB that interfaces an array of sensors to the Arduino Mega 2560 controller to acquire pollutant and meteorological parameters. These data are wirelessly transmitted to the Smart Fog Environmental Gateway (SFEG) in the fog layer using LoRa, a long-range, low-cost, and low-power LPWAN-based solution. Further, an efficient Fog Intelligence is facilitated in the SFEG through an optimized lightweight DL-based Seq2Seq GRU Attention model for real-time accurate forecasting, timely early warnings, and faster event response with effective utilization of fog resources. The Seq2Seq GRU Attention model for multivariate multi-step forecasting of air quality levels combines the strength of the Seq2Seq architecture with the attention mechanism. The experimental results reveal that Seq2Seq GRU Attention improved the forecasting accuracy and stability in comparison with the state-of-the-art DL methods, with an average RMSE of 5.5576, MAE of 3.4975, MAPE of 19.1991, R^2^ of 0.6926, and Theil’s U1 of 0.1325 across the six primary pollutants (PM_2.5_, CO_2_, CO, SO_2_, NO_2_, and O_3_) for the future twelve time steps. Subsequently, to facilitate optimized model deployment on the resource-constrained fog nodes like SFEG, post-training quantization techniques like dynamic range quantization, integer with float fallback quantization, full-integer-only quantization, float16 quantization, are applied to the initial model. The evaluation shows that the results achieved through dynamic range quantization outperformed other methods with a significant reduction in file size, good forecast accuracy, and improved execution time, striking a balance between performance and computational efficiency. This makes the proposed optimized lightweight model suitable for deployment on the SFEG to enable efficient Fog Intelligence and thereby enhance the effectiveness of the FAQMP system. Furthermore, the EnviroWeb application presents real-time air quality data and alerts to the stakeholders. In summary, the proposed FAQMP system enables real-time and low-cost monitoring, efficient communication, accurate multi-step forecasting with optimized fog resource utilization, and decision support through timely early warnings, and event responses. This will empower informed decision making to address pollution concerns and maintain safe AQI levels in smart cities

While intelligence in Fog Computing is still in its early stages, our study offers promising directions for further exploration. Despite its advantages, a limitation of this work is that the performance of the proposed optimized lightweight model has not been analyzed across various resource-constrained fog environments. Future work will address this by evaluating the model in different settings. Additionally, the accuracy of sensor data collected in the AQM node will be enhanced through automated calibration procedures to improve reliability and consistency in air quality measurements.

## Figures and Tables

**Figure 1 sensors-24-05069-f001:**
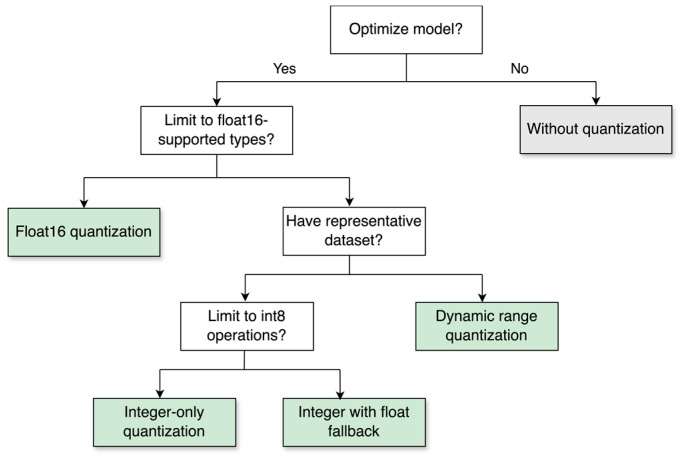
Post-training optimization methods provided by TensorFlow.

**Figure 2 sensors-24-05069-f002:**
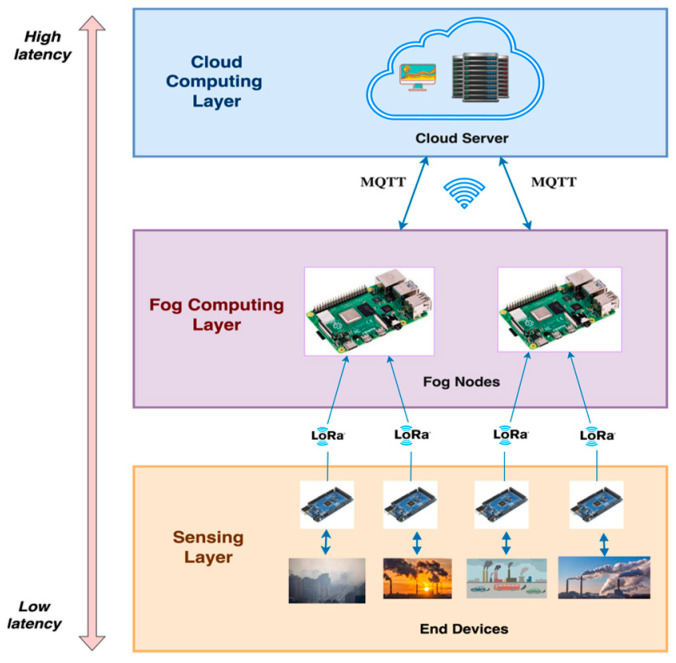
A three-layered Fog Computing-based architecture of the proposed system.

**Figure 3 sensors-24-05069-f003:**
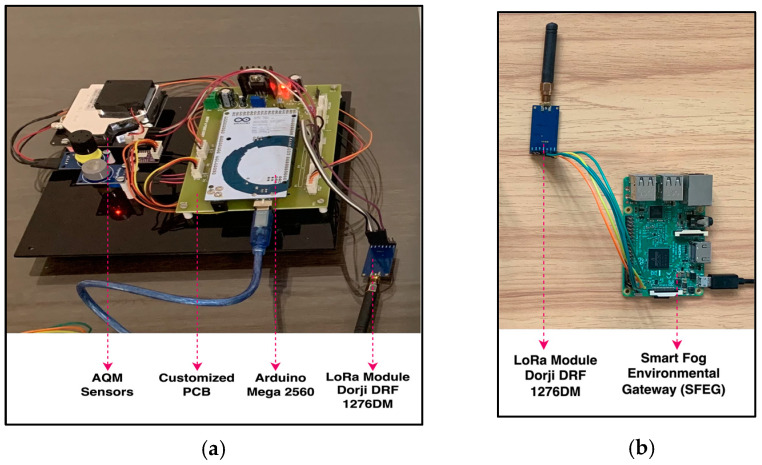
Hardware of the proposed FAQMP system. (**a**) Air Quality Monitoring (AQM) Sensor Node. (**b**) Smart Fog Environmental Gateway (SFEG).

**Figure 4 sensors-24-05069-f004:**
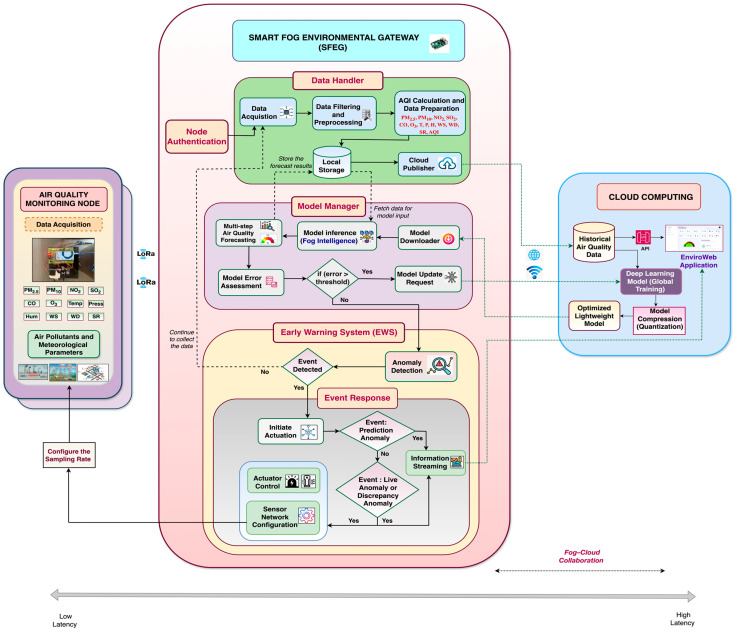
Architecture and data flow of the proposed Fog-enabled Air Quality Monitoring and Prediction (FAQMP) System.

**Figure 5 sensors-24-05069-f005:**
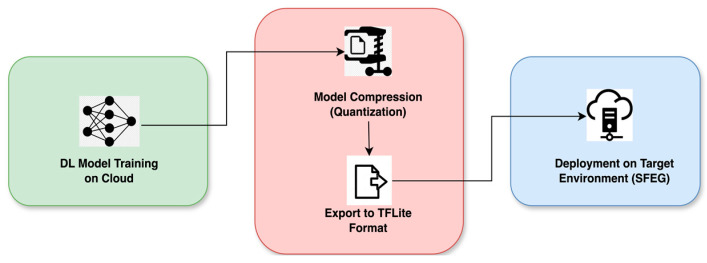
DL model deployment pipeline after model quantization.

**Figure 6 sensors-24-05069-f006:**
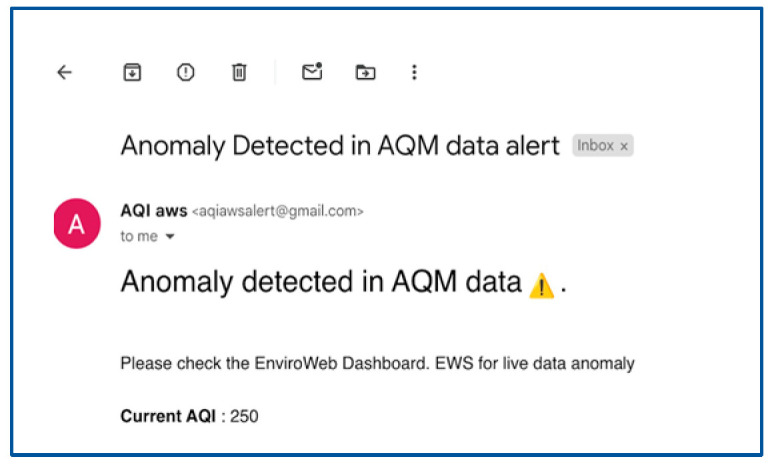
Real-time alerts triggered by anomalous AQI Levels via email.

**Figure 7 sensors-24-05069-f007:**
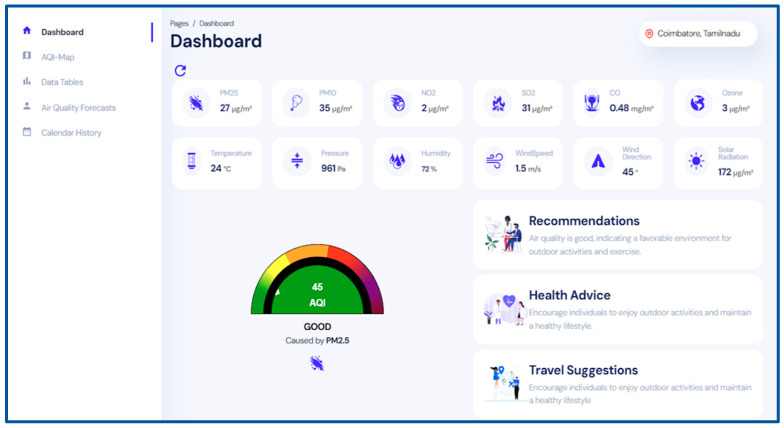
Graphical User Interface of the EnviroWeb application displaying the live pollutants, Air Quality Index (AQI) level, and recommendations for citizens in real time.

**Figure 8 sensors-24-05069-f008:**
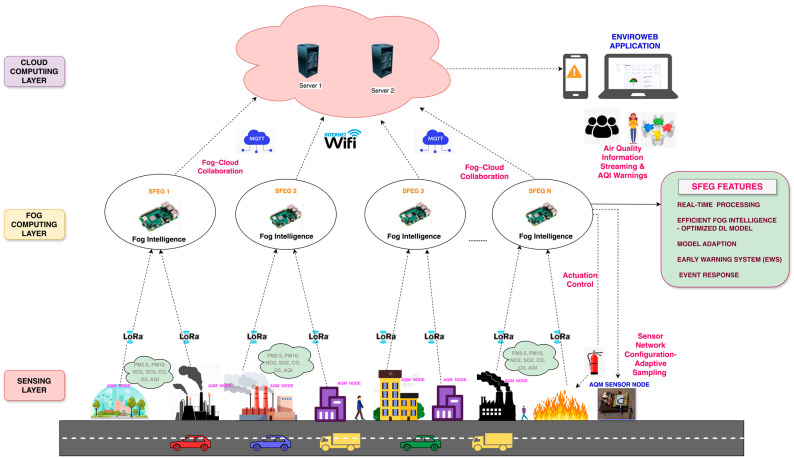
City-wide implementation of the proposed FAQMP system.

**Figure 9 sensors-24-05069-f009:**
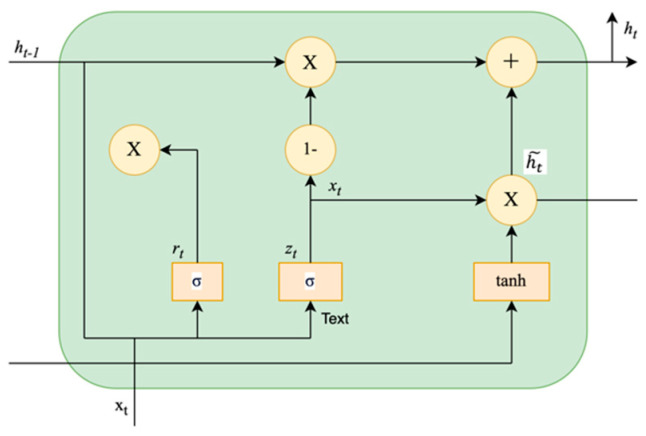
GRU architecture.

**Figure 10 sensors-24-05069-f010:**
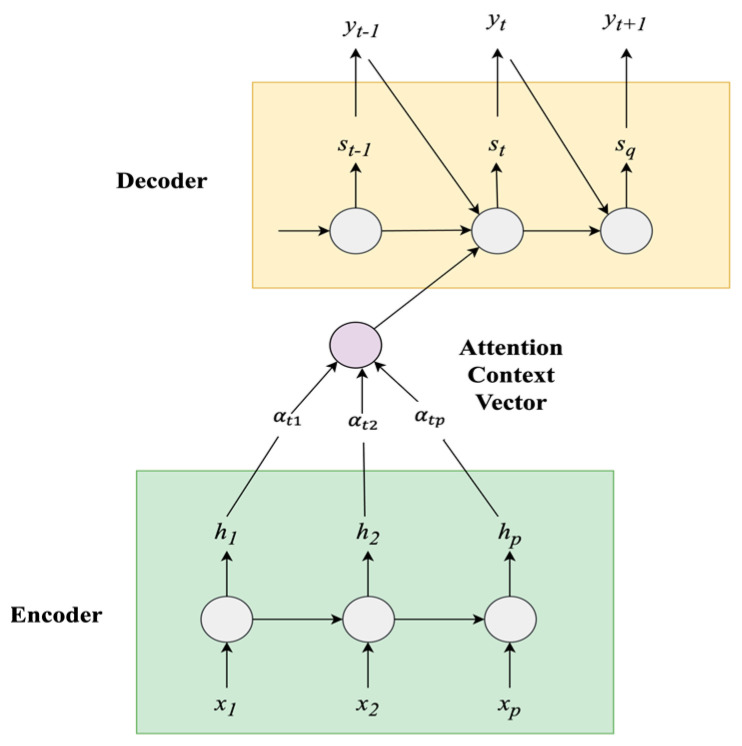
Architecture of the Sequence-to-Sequence GRU Attention mechanism.

**Figure 11 sensors-24-05069-f011:**
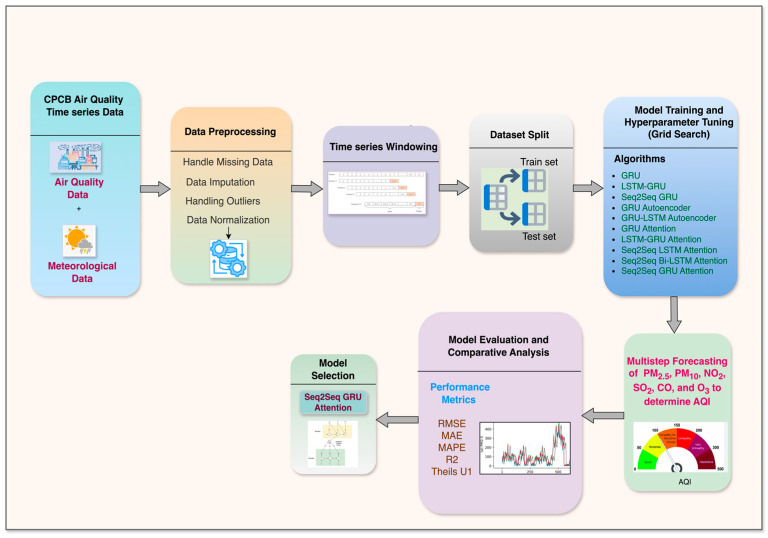
Steps involved in multivariate multi-step air quality forecasting.

**Figure 12 sensors-24-05069-f012:**
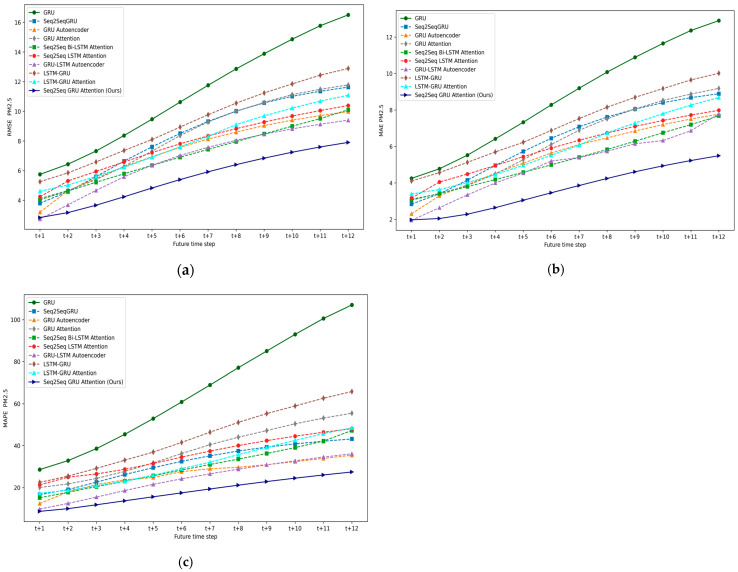
Error metrics of DL models to forecast PM_2.5_ over twelve time steps (t1–t12). (**a**) RMSE comparison; (**b**) MAE comparison; (**c**) MAPE comparison.

**Figure 13 sensors-24-05069-f013:**
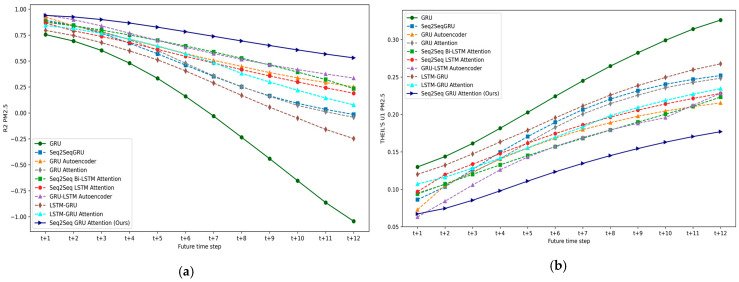
Performance metrics of DL models to forecast PM_2.5_ over twelve time steps (t1–t12). (**a**) R^2^ comparison; (**b**) Theil’s U1 comparison.

**Figure 14 sensors-24-05069-f014:**
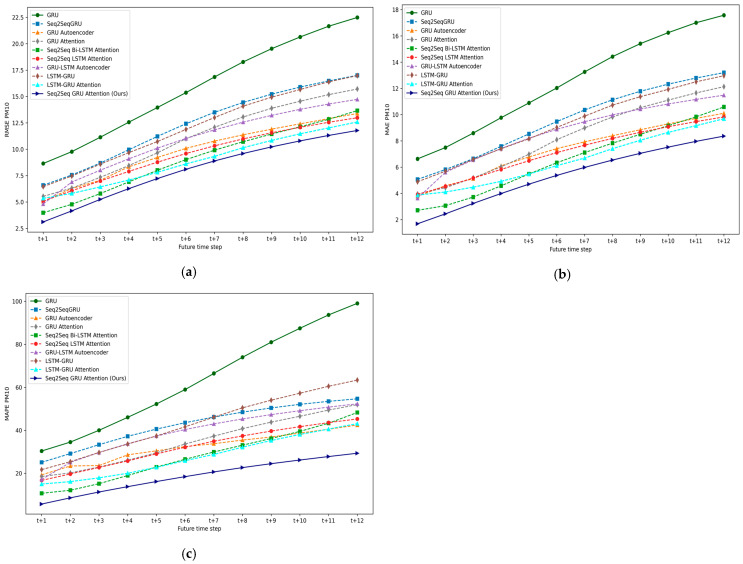
Error metrics of DL models to forecast PM_10_ over twelve time steps (t1–t12). (**a**) RMSE comparison; (**b**) MAE comparison; (**c**) MAPE comparison.

**Figure 15 sensors-24-05069-f015:**
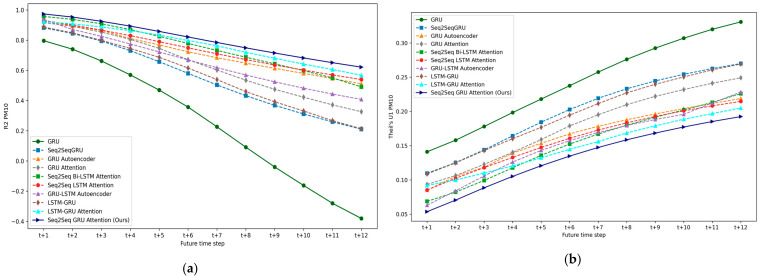
Performance metrics of DL models to forecast PM_10_ over twelve time steps (t1–t12). (**a)** R^2^ comparison; (**b**) Theil’s U1 comparison.

**Figure 16 sensors-24-05069-f016:**
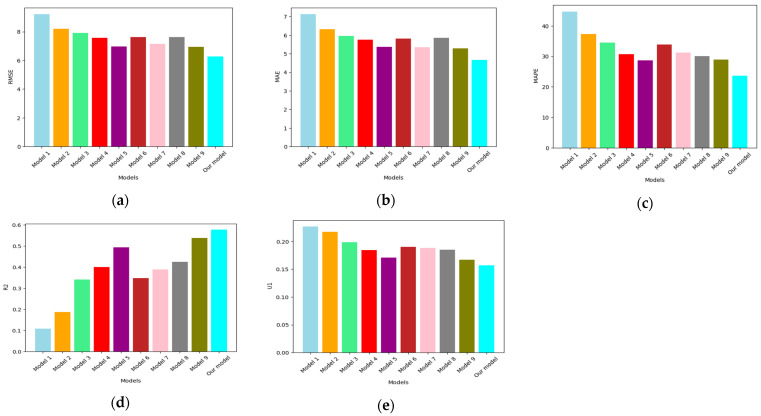
Performance metrics (RMSE, MAE, MAPE, R^2^, and U1) of the compared models across all pollutants (PM2.5, PM10, NO2, SO2, CO, and O3) over 12 time steps (t1–t12): (**a**) Average RMSE; (**b**) Average MAE; (**c**) Average MAPE; (**d**) Average R^2^; (**e**) Average Theil’s U1; and Model 1—GRU, Model 2—LSTM-GRU, Model 3—Seq2Seq GRU, Model 4—GRU Autoencoder, Model 5—GRU-LSTM Autoencoder, Model 6—GRU Attention, Model 7—LSTM-GRU Attention, Model 8—Seq2Seq LSTM Attention, Model 9—Seq2Seq Bi-LSTM Attention, and Our model—Seq2Seq GRU Attention.

**Figure 17 sensors-24-05069-f017:**
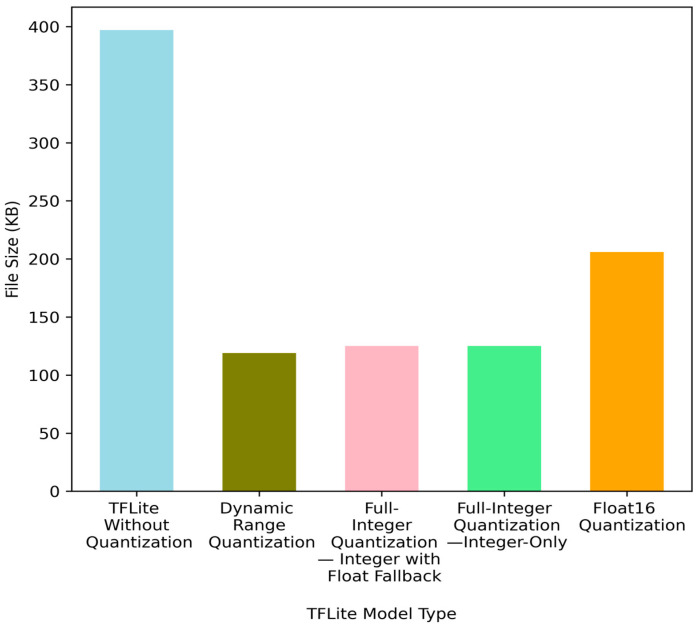
TensorFlow Lite models—file size comparison.

**Table 1 sensors-24-05069-t001:** AQI remarks by CPCB in India.

AQI	Descriptor	Indicative Color	Associated Health Impacts
0 to 50	Good	Green	Poses minimal or no risk
51 to 100	Satisfactory	Yellow	Acceptable air quality level. Minor concern for the sensitive members of the population
101 to 200	Moderately Polluted	Orange	Sensitive members of the population may experience health effects from prolonged exposure.
201 to 300	Poor	Red	The public may start to experience illness. Severe effect on the sensitive members of the population.
301 to 400	Very poor	Purple	Health alert. Serious health impacts for everyone.
401 to 500	Severe	Maroon	Emergency warning. Everyone is likely to be affected.

**Table 2 sensors-24-05069-t002:** Summary of the existing AQM systems.

Reference	Pollutants Monitored	Controllers, Sensors, and Other Hardware Platforms	Meteorological Parameters	Edge/Fog Computing	LPWAN	Cloud Computing	Air Quality Forecasting	Optimized DL Model-Fog Intelligence	Early Warnings	Web Application
Laskar et al. [22]	CO, CO_2_, PM_2.5_, PM_10_, SO_2_, NO_2_	ESP8266, MQ2, MQ 7, MQ131, MQ135, MQ 136	X	X	X	√	X	X	X	√
Kumar et al. [25]	PM_10_, PM_2.5_, NO_2_, CO_2_	Intel Edison, GPS, MQ131, MQ135, MQ136, MQ7 sensors	X	X	X	√	√	X	√	√
Aashiq et al. [28]	PM_2.5_, PM_10_, VOC, CO, Temperature, Pressure, Humidity, Altitude	Arduino Uno, ESP 8266, BMP280, AHT10, MQ7	√	X	X	√	X	X	X	√
Asha et al. [29]	PM_2.5_, CO, CO_2_, NH_3_, NO_2_, CH_4_, Temperature, Humidity	Grove-Multichannel Gas Sensor, MHZ19, DHT11, HM3301 laser, PM2.5 sensor	√	X	X	√	√	X	√	X
Moses et al. [39]	PM_2.5_, PM_10_, CO, O_3_, NO_2_, SO_2_.	MQ136, MQ7, RPi 3, PM_2.5_ sensor, NO_2_ sensor, O_3_ sensor, NB-IoT.	X	X	√	√	√	X	X	√
Lai X et al. [33]	PM_2.5_, PM_10_, SO_2_, NO_2_, CO, O_3_	ESP8266, Raspberry Pi 3 Model B, ZH03A, SGA-700 Intelligent Gas Sensor	X	√	X	√	√	X	X	√
Senthilkumar et al. [34]	PM_2.5_, PM_10_, SO_2_, NO_2_, CO, O_3_	GP2Y1014AU0F, GSNT11, DSM501, MQ-7, SO2-AF, MiCS2610-11, DHT11.	X	√	X	√	√	X	X	√
Moursi et al. [35]	PM_2.5_, CO, CO_2_, Temperature, Pressure, Wind Speed	Node MCU, MQ7, MQ135, RPi 4	√	√	X	√	√	X	X	X
Santos et al. [36]	PM_1_, PM_2.5_, PM_10_.	PM sensor, Lora WAN, Sigfox, DASH7	X	√	√	√	X	X	√	√
Kristiani et al. [37]	PM_1.0_, PM_2.5_, PM10	GlobalSat LM-130, Arduino Uno, Raspberry Pi 3, PMS5003T G5T	X	√	√	√	X	X	X	√
Jabbar et al. [38]	NO_2_, SO_2_, CO_2_, CO, PM_2.5_, Temperature, Humidity	Arduino Uno, MQ 135, MQ 9, PMS3003, MQ 136, MiCS-4514, DHT11	√	X	√	√	X	X	X	√
Proposed FAQMP system	PM_2.5_, PM_10_, NO_2_, SO_2_, CO, O_3_, Wind Speed, Wind Direction, Temperature, Pressure, Humidity,	Arduino Mega 2560, RPi 3 Model B, SDS011, MICS-4514, MQ7, MQ136, BME280, MQ131, NEO-6M, PXV7002DP, DRF1276DM,	√	√	√	√	√	√	√	√

√: Considered by the system, X: Not considered by the system.

**Table 3 sensors-24-05069-t003:** Comparison of hardware platforms for Fog Computing with their specifications.

Device	CPU	GPU	RAM	Flash Memory	Power Consumption	GPO	Interfaces for external sensors and actuators	Supported framework	Cost (INR)
Raspberry Pi 4 Model B	4-Core ARM Cortex A72	Broadcom Video Core VI	2 GB, 4 GB, or 8 GB LPDDR4 RAM	MicroSD card slot	2.7–7 W	40 pins	Bluetooth 5.0 Gigabit Ethernet, Wi-Fi (802.11ac),	TensorFlow Lite, PyTorch, OpenCV, MXNet, Keras, OpenCV	5300
Raspberry Pi 3 Model B +	4-core ARM Cortex A53.	Broadcom Video Core IV	1 GB LPDDR2-900 SDRAM (32-bit)	8 GB eMMC MicroSD card slot	2.5–4 W	40 pins	10/100 Ethernet, Wi-Fi (802.11ac), Bluetooth 4.2	TensorFlow Lite, PyTorch, OpenCV, MXNet, Keras, OpenCV	3700
Jetson TX2	Dual-Core NVIDIA Denver 2 + Quad-Core ARM Cortex A57 MP Core	NVIDIA 256 CUDA Cores (Pascal GPU)	8 GB LPDDR4 (128-bit)	32 GB eMMC, SDIO, SATA	7.5–15 W	12 pins	802.11a/b/g/n/ac 2 × 2 867 Mbps, Bluetooth 4.1, 10/100/1000 BASE-T Ethernet	TensorFlow, PyTorch, Caffe, MXNet, and other prominent frameworks	16,500
Jetson Nano	4-Core ARM Cortex-A57 MP Core	NVIDIA 128 CUDA cores (Maxwell GPU)	8 GB LPDDR4 (64-bit)	16 GB eMMC	5–10 W	40 pins	Gigabit Ethernet	TensorFlow, PyTorch, Caffe, MXNet, CoreML, TensorRT, Keras, and others	23,000
Beagle Boneblack AI-64	TI TDA4VM ARMCortex-A72(64 bit) processor	SGX544	4 GB LPDDR4	16 GB eMMC	12.5 W	96 pins	Gigabit Ethernet	TensorFlow, PyTorch, Caffe, OpenCV	15,000
NVIDIA Jetson AGX Xavier	8-core ARM Cortex-A57 (64-bit)	512 CUDA Cores with 64 Tensor Cores (Volta GPU)	32 GB LPDDR4x (256-bit)	32 GB eMMC	10–30 W	160 pins	Wi-Fi	TensorFlow, PyTorch, TensorRT, OpenCV, cuDNN	120,300
Google Coral Dev Board	4-Core ARM Cortex-A53, Cortex-M4F	Integrated GC7000 Lite Graphics	1 or 4 GB LPDDR4	8 or 16 GB eMMC	5–10 W	40 pins	Wi-Fi 2 × 2 MIMO (802.11b/g/n/ac 2.4/5 GHz) and Bluetooth 4.2	TensorFlow, PyTorch, OpenCV	19,000
ODYSSEY—X86J4125800 v2	Intel Celeron J4105, Quad-Core	Intel UHD Graphics 600.	LPDDR4 8 GB	64 GB eMMC V5.1	6–10 W	40 pins	Wi-Fi (x), Bluetooth (BLE 5.0).	TensorFlow, PyTorch, OpenCV	34,000

**Table 4 sensors-24-05069-t004:** Sensors to monitor the pollutants and meteorological parameters.

Serial No.	Parameters	Sensor	Unit
1	Particulate Matter 2.5 (PM_2.5_) and Particulate Matter 10 (PM_10_)	SDS011	µg/m^3^
2	Nitrogen Dioxide (NO_2_) and Carbon Monoxide (CO)	MICS-4514	µg/m^3^
3	Sulfur Dioxide (SO_2_)	MQ-136	µg/m^3^
6	Ozone (O_3_)	MQ-131	µg/m^3^
7	Ambient Temperature	BME280	°C
8	Relative Humidity	BME280	%
9	Pressure	BME280	hPa
10	Solar Radiation (SR)	Pyranometer	W/m^2^
11	Wind Speed (WS)	MPXV7002DP	m/s
12	Wind Direction (WD)	MPXV7002DP	Degrees

**Table 5 sensors-24-05069-t005:** A summary of the tools and technologies in the FAQMP system.

Layers/Services	Tools/Technologies/Methods
Sensing Layer	Arduino Mega 2560, Customized PCB, MICS-4514, SDS011, MQ136, BME280, and MQ131
Communication Layer	LoRa-DRF1276DM
Fog Computing Layer	Raspberry Pi Model 3B+
Air Quality Forecasting	Seq2Seq-GRU Attention model
Optimized Lightweight DL Model	Dynamic Range Quantization
Application Layer	MQTT
Cloud Computing Layer	AWS IoT Core, AWS Dynamo DB, AWS Lambda, AWS Sage Maker, and AWS S3

**Table 6 sensors-24-05069-t006:** Hyperparameters of the proposed Seq2Seq GRU Attention model and the optimal values.

Hyperparameters	Range of Values	Optimal Value
Encoder GRU layers	[1, 2, 3]	1
Decoder GRU layers	[1, 2, 3]	1
No. of units in encoder	[32, 64,128, 256]	128
No. of units in decoder	[32, 64, 128, 256]	128
Activation function in encoder	[‘relu’, ‘tanh’, ‘sigmoid’]	tanh
Activation function in decoder	[‘relu’, ‘tanh’, ‘sigmoid’]	tanh
Optimizer	[‘adam’, ‘rmsprop’, ‘sgd’]	adam
Learning rate	[0.001, 0.01, 0.1]	0.001
Batch size	[32, 64, 128, 256, 512]	256
Epochs	[20, 50, 100, 200]	100
Window size	[97, 194, 292]	194
Attention mechanism	[‘additive’, ‘multiplicative’]	additive

**Table 7 sensors-24-05069-t007:** Evaluation of the DL-based multivariate multi-step forecasting models to forecast PM_2.5_ for 12 consecutive time steps.

Metrics	Models	t + 1	t + 2	t + 3	t + 4	t + 5	t + 6	t + 7	t + 8	t + 9	t + 10	t + 11	t + 12
RMSE	GRU	5.7503	6.4391	7.3231	8.3711	9.472	10.625	11.7603	12.8639	13.8863	14.865	15.7721	16.5042
	Seq2Seq GRU	3.8051	4.5847	5.6031	6.6462	7.6098	8.5108	9.3244	10.0155	10.5739	11.0134	11.3557	11.6317
	GRU Autoencoder	3.205	4.6504	5.4774	6.2722	6.9464	7.5749	8.1332	8.599	9.0422	9.4081	9.7157	9.9815
	GRU Attention	4.1281	4.6628	5.3978	6.2827	7.321	8.3666	9.2861	10.0054	10.6013	11.1322	11.4877	11.7839
	Seq2Seq Bi-LSTM Attention	4.0414	4.6366	5.2174	5.7907	6.3591	6.9076	7.4375	7.961	8.4845	9.0014	9.5115	10.1029
	Seq2Seq LSTM Attention	4.2439	5.3086	5.9508	6.5895	7.2215	7.8119	8.3474	8.835	9.2789	9.6827	10.0529	10.3945
	GRU-LSTM Autoencoder	2.7411	3.6969	4.6703	5.588	6.3667	7.0247	7.5778	8.0521	8.4631	8.8208	9.1321	9.4059
	LSTM-GRU	5.2559	5.8554	6.5909	7.3631	8.1011	8.9407	9.7744	10.5525	11.2438	11.8507	12.437	12.8935
	LSTM-GRU Attention	4.6106	5.0389	5.6083	6.2303	6.9063	7.6123	8.3108	9.1083	9.695	10.2171	10.6866	11.086
	Seq2Seq GRU Attention (Ours)	**2.8438**	**3.1747**	**3.6697**	**4.2402**	**4.8334**	**5.4005**	**5.9256**	**6.4073**	**6.8464**	**7.2452**	**7.5991**	**7.9083**
MAE	GRU	4.2502	4.7761	5.5214	6.4163	7.3286	8.2804	9.1957	10.0801	10.8906	11.6531	12.3603	12.9039
	Seq2Seq GRU	2.8393	3.4075	4.1512	4.9589	5.7292	6.4473	7.0811	7.6153	8.0533	8.4036	8.679	8.8931
	GRU Autoencoder	2.3058	3.2937	3.9066	4.5351	5.085	5.6391	6.1092	6.47	6.8515	7.1996	7.5064	7.7691
	GRU Attention	3.1157	3.4011	3.8623	4.489	5.2756	6.1126	6.8874	7.5178	8.0548	8.5264	8.8737	9.1849
	Seq2Seq Bi-LSTM Attention	3.0571	3.4225	3.7978	4.1828	4.5842	4.9925	5.4079	5.8336	6.2902	6.7529	7.1971	7.6832
	Seq2Seq LSTM Attention	3.1687	4.0508	4.4868	4.9556	5.4401	5.9089	6.3417	6.7414	7.1019	7.4269	7.7209	7.9858
	GRU-LSTM Autoencoder	1.9516	2.6403	3.3455	3.9976	4.5428	5.2029	5.3922	5.7365	6.1489	6.3249	6.8662	7.7834
	LSTM-GRU	4.1088	4.5652	5.1299	5.6994	6.2299	6.8745	7.5312	8.1527	8.6954	9.1747	9.6509	10.0171
	LSTM-GRU Attention	3.3968	3.6374	4.0196	4.4594	4.9671	5.5189	6.0747	6.7468	7.2903	7.7982	8.2698	8.6823
	Seq2Seq GRU Attention (Ours)	**1.9761**	**2.0486**	**2.2852**	**2.6434**	**3.0538**	**3.4597**	**3.8588**	**4.2456**	**4.6125**	**4.9381**	**5.2296**	**5.4929**
MAPE	GRU	28.6139	32.9009	38.6036	45.4484	52.9217	60.8581	68.9421	77.1333	85.0608	93.006	100.5752	107.0597
	Seq2Seq GRU	16.6686	19.0712	22.6128	26.2669	29.5448	32.5289	35.1817	37.4681	39.3389	40.877	42.1503	43.2279
	GRU Autoencoder	12.4442	17.946	21.4561	23.738	24.8583	27.6232	28.9988	29.7139	30.9402	32.3991	33.9318	35.4523
	GRU Attention	20.0478	21.7856	24.3714	27.7212	31.798	36.2378	40.4969	44.0495	47.2186	50.4469	53.1723	55.4954
	Seq2Seq Bi-LSTM Attention	15.158	17.8068	20.4797	23.0683	25.6571	28.3767	31.0354	33.6213	36.2953	39.082	42.2267	47.253
	Seq2Seq LSTM Attention	21.3452	25.0016	26.5273	28.7542	31.5503	34.5531	37.4244	40.0693	42.4354	44.5397	46.4283	48.1298
	GRU-LSTM Autoencoder	9.7933	12.4743	15.4891	18.6239	21.5598	24.2152	26.5969	28.8085	30.8915	32.8082	34.5628	36.1578
	LSTM-GRU	22.516	25.5122	29.1859	33.1051	36.9586	41.5559	46.4527	51.1122	55.328	58.9335	62.6408	65.826
	LSTM-GRU Attention	17.309	18.7261	20.814	23.1675	25.9111	29.0349	32.1423	35.7822	39.1718	42.5398	45.7723	48.5527
	Seq2Seq GRU Attention (Ours)	**8.6976**	**9.9819**	**11.8298**	**13.7335**	**15.6353**	**17.5067**	**19.3686**	**21.1848**	**22.9085**	**24.5232**	**26.0593**	**27.4658**
R^2^	GRU	0.7552	0.6928	0.6022	0.4797	0.333	0.1598	−0.0306	−0.2348	−0.4407	−0.653	−0.8634	−1.0432
	Seq2Seq GRU	0.8928	0.8443	0.7671	0.672	0.5695	0.4609	0.3521	0.2515	0.1646	0.0926	0.0341	−0.0149
	GRU Autoencoder	0.924	0.8398	0.7775	0.7079	0.6413	0.5729	0.5071	0.4483	0.3891	0.3379	0.2929	0.2526
	GRU Attention	0.8738	0.8389	0.7839	0.7069	0.6016	0.479	0.3574	0.253	0.1603	0.073	0.0115	−0.0416
	Seq2Seq Bi-LSTM Attention	0.8791	0.8407	0.7981	0.751	0.6994	0.6449	0.5878	0.5271	0.4622	0.3939	0.3223	0.2344
	Seq2Seq LSTM Attention	0.8667	0.7912	0.7373	0.6776	0.6123	0.5458	0.4808	0.4176	0.3567	0.2987	0.243	0.1895
	GRU-LSTM Autoencoder	0.9444	0.8987	0.8382	0.7681	0.6987	0.6327	0.5721	0.5162	0.4649	0.418	0.3753	0.3364
	LSTM-GRU	0.7955	0.746	0.6778	0.5974	0.5121	0.4051	0.2881	0.1691	0.0554	−0.0506	−0.1587	−0.247
	LSTM-GRU Attention	0.8426	0.8119	0.7667	0.7118	0.6454	0.5687	0.4853	0.381	0.2977	0.2191	0.1445	0.0781
	Seq2Seq GRU Attention (Ours)	**0.9401**	**0.9253**	**0.9001**	**0.8665**	**0.8263**	**0.7829**	**0.7383**	**0.6937**	**0.6498**	**0.6073**	**0.5674**	**0.5309**
Theil’s U1	GRU	0.1299	0.1438	0.1613	0.1816	0.2028	0.2243	0.245	0.2646	0.2824	0.2991	0.3141	0.3261
	Seq2Seq GRU	0.0862	0.1035	0.1265	0.1496	0.1705	0.1896	0.2065	0.2206	0.2317	0.2403	0.2469	0.252
	GRU Autoencoder	0.0728	0.1051	0.1234	0.1405	0.1551	0.1683	0.1798	0.1891	0.1977	0.2046	0.2105	0.2155
	GRU Attention	0.0951	0.1067	0.1224	0.141	0.1621	0.1828	0.2007	0.2145	0.2258	0.2358	0.2428	0.2484
	Seq2Seq Bi-LSTM Attention	0.0938	0.1071	0.12	0.1327	0.1451	0.1569	0.1682	0.1791	0.1898	0.2004	0.2109	0.2233
	Seq2Seq LSTM Attention	0.0969	0.1197	0.1337	0.1478	0.1617	0.1745	0.1861	0.1965	0.2058	0.2141	0.2214	0.2281
	GRU-LSTM Autoencoder	0.0631	0.0841	0.1058	0.1261	0.1431	0.1574	0.1693	0.1794	0.1882	0.1958	0.2124	0.2283
	LSTM-GRU	0.12	0.1323	0.1473	0.1633	0.1788	0.1955	0.2114	0.2259	0.2386	0.2495	0.2597	0.2677
	LSTM-GRU Attention	0.1071	0.1162	0.1284	0.1417	0.1555	0.1697	0.1834	0.1984	0.2093	0.2188	0.2271	0.2345
	Seq2Seq GRU Attention (Ours)	**0.0672**	**0.0745**	**0.0854**	**0.098**	**0.111**	**0.1234**	**0.1347**	**0.1451**	**0.1545**	**0.163**	**0.1705**	**0.177**

The bold-font numbers represent the data in this study.

**Table 8 sensors-24-05069-t008:** Evaluation of the DL-based multivariate multi-step forecasting models to forecast PM_10_ for 12 consecutive time steps.

Metrics	Models	t + 1	t + 2	t + 3	t + 4	t + 5	t + 6	t + 7	t + 8	t + 9	t + 10	t + 11	t + 12
RMSE	GRU	8.6494	9.7725	11.1405	12.5685	13.9573	15.3674	16.8543	18.2741	19.5324	20.6441	21.6628	22.4912
	Seq2Seq GRU	6.589	7.5597	8.6918	9.9564	11.2206	12.4176	13.5015	14.4357	15.2224	15.8943	16.4795	17.0052
	GRU Autoencoder	5.0651	6.2794	7.0571	8.3398	9.2232	10.0855	10.7712	11.3692	11.9072	12.4118	12.9025	13.388
	GRU Attention	5.5334	6.3184	7.3395	8.4557	9.662	11.001	12.0816	13.0618	13.8828	14.5538	15.178	15.7099
	Seq2Seq Bi-LSTM Attention	3.9835	4.782	5.7994	6.9122	8.0085	9.016	9.9137	10.7138	11.4445	12.1333	12.8444	13.6602
	Seq2Seq LSTM Attention	5.0743	6.0668	7.0002	7.9013	8.7771	9.5945	10.3264	10.9808	11.5641	12.0851	12.5569	12.99
	GRU-LSTM Autoencoder	4.8163	6.8946	8.0143	9.0939	10.1088	11.0224	11.8404	12.5679	13.2147	13.7854	14.2871	14.7291
	LSTM-GRU	6.4579	7.4615	8.591	9.6816	10.7287	11.8831	13.0104	14.084	14.9369	15.6643	16.391	16.9778
	LSTM-GRU Attention	5.346	5.8164	6.433	7.0836	7.8192	8.6162	9.3259	10.15	10.834	11.4686	12.0238	12.5863
	Seq2Seq GRU Attention (Ours)	**3.1161**	**4.1579**	**5.251**	**6.2687**	**7.2239**	**8.1036**	**8.8936**	**9.598**	**10.227**	**10.7951**	**11.3114**	**11.7805**
MAE	GRU	6.6286	7.4907	8.5942	9.7669	10.8913	12.0344	13.2573	14.4222	15.4112	16.2477	17.0041	17.5762
	Seq2Seq GRU	5.0652	5.8155	6.6415	7.5874	8.5311	9.4732	10.357	11.1315	11.7817	12.3253	12.7906	13.2017
	GRU Autoencoder	3.7739	4.5835	5.1161	6.1029	6.7761	7.4096	7.9304	8.3951	8.8504	9.2977	9.7143	10.1065
	GRU Attention	3.9513	4.4332	5.1776	6.0242	6.9797	8.0952	8.9907	9.8208	10.5319	11.1087	11.6561	12.1304
	Seq2Seq Bi-LSTM Attention	2.7099	3.0622	3.7211	4.5768	5.4795	6.3327	7.1086	7.8256	8.5035	9.1589	9.8334	10.587
	Seq2Seq LSTM Attention	3.9146	4.5345	5.166	5.8244	6.4891	7.1105	7.6726	8.1881	8.6554	9.0811	9.4708	9.8271
	GRU-LSTM Autoencoder	3.6318	5.6068	6.5531	7.4168	8.2029	8.8831	9.4647	9.9818	10.4253	10.8131	11.1613	11.4833
	LSTM-GRU	4.8847	5.6879	6.5941	7.4009	8.1655	9.0283	9.8853	10.7254	11.3719	11.9254	12.5039	12.965
	LSTM-GRU Attention	3.8687	4.104	4.4724	4.9181	5.4713	6.1136	6.6916	7.4075	8.0468	8.6485	9.1599	9.689
	Seq2Seq GRU Attention (Ours)	**1.6846**	**2.4383**	**3.2357**	**3.9901**	**4.7036**	**5.3709**	**5.9807**	**6.5387**	**7.0537**	**7.5246**	**7.9651**	**8.3638**
MAPE	GRU	30.3727	34.518	40.0164	46.0585	52.2646	58.999	66.4663	73.9537	80.9814	87.4455	93.671	99.0517
	Seq2Seq GRU	25.0938	29.1666	33.3131	37.2118	40.5834	43.554	46.1849	48.4692	50.4316	52.1105	53.4974	54.6935
	GRU Autoencoder	19.2579	23.3933	23.5905	28.5973	30.4427	32.4252	33.7576	35.4731	36.8835	38.7165	40.5186	42.4642
	GRU Attention	18.5097	20.1715	22.92	26.1081	29.5136	33.6202	37.283	40.794	43.8232	46.5455	49.4513	51.9639
	Seq2Seq Bi-LSTM Attention	10.6661	12.0867	15.1548	18.9536	22.879	26.5145	29.9031	33.0783	36.2101	39.5101	43.3289	48.2951
	Seq2Seq LSTM Attention	16.6997	19.6764	22.7178	25.741	29.0354	32.1313	34.9145	37.4047	39.6579	41.7055	43.5818	45.3093
	GRU-LSTM Autoencoder	17.0974	25.0763	29.6866	33.7503	37.343	40.3772	43.0011	45.3059	47.3129	49.1332	50.8132	52.3643
	LSTM-GRU	21.6967	25.4013	29.6879	33.6314	37.3767	41.6565	46.1184	50.4935	54.0316	57.2879	60.5166	63.4052
	LSTM-GRU Attention	14.9829	16.1077	17.8634	20.0067	22.7702	25.8781	28.7723	32.1504	35.2646	38.0895	40.5817	43.1119
	Seq2Seq GRU Attention (Ours)	**5.6667**	**8.5543**	**11.2846**	**13.7689**	**16.1524**	**18.4329**	**20.6252**	**22.6466**	**24.4721**	**26.1693**	**27.79**	**29.3231**
R^2^	GRU	0.7963	0.74	0.6621	0.5698	0.4694	0.3567	0.2261	0.091	−0.04	−0.1622	−0.2803	−0.3807
	Seq2Seq GRU	0.8818	0.8444	0.7943	0.7301	0.6571	0.58	0.5034	0.4321	0.3683	0.3111	0.2591	0.2107
	GRU Autoencoder	0.9302	0.8926	0.8644	0.8106	0.7683	0.7229	0.6839	0.6478	0.6135	0.5799	0.5458	0.5108
	GRU Attention	0.9166	0.8913	0.8533	0.8053	0.7457	0.6704	0.6023	0.5351	0.4746	0.4224	0.3715	0.3264
	Seq2Seq Bi-LSTM Attention	0.9568	0.9377	0.9084	0.8699	0.8253	0.7786	0.7322	0.6872	0.643	0.5985	0.5499	0.4907
	Seq2Seq LSTM Attention	0.9299	0.8998	0.8666	0.83	0.7902	0.7492	0.7095	0.6714	0.6355	0.6017	0.5698	0.5394
	GRU-LSTM Autoencoder	0.9368	0.8706	0.8251	0.7748	0.7217	0.6691	0.618	0.5696	0.524	0.4818	0.4431	0.4079
	LSTM-GRU	0.8865	0.8484	0.799	0.7448	0.6865	0.6154	0.5388	0.4594	0.3918	0.3309	0.267	0.2133
	LSTM-GRU Attention	0.9222	0.9079	0.8873	0.8634	0.8335	0.7978	0.763	0.7192	0.68	0.6413	0.6056	0.5676
	Seq2Seq GRU Attention (Ours)	**0.9736**	**0.9529**	**0.9249**	**0.893**	**0.8579**	**0.8211**	**0.7845**	**0.749**	**0.7149**	**0.6822**	**0.6509**	**0.6212**
Theil’s U1	GRU	0.1411	0.158	0.178	0.1984	0.2181	0.2376	0.2575	0.2761	0.2925	0.307	0.32	0.3308
	Seq2Seq GRU	0.1097	0.1253	0.1439	0.1643	0.1843	0.2028	0.2192	0.2331	0.2446	0.2543	0.2626	0.27
	GRU Autoencoder	0.085	0.1052	0.1185	0.1392	0.1535	0.1673	0.1783	0.1879	0.1964	0.2042	0.2117	0.2189
	GRU Attention	0.0935	0.1063	0.1227	0.1403	0.1589	0.1789	0.1952	0.2099	0.222	0.232	0.2412	0.2491
	Seq2Seq Bi-LSTM Attention	0.0687	0.0822	0.0992	0.1177	0.1359	0.1524	0.167	0.1799	0.1915	0.2023	0.2132	0.226
	Seq2Seq LSTM Attention	0.0853	0.1023	0.118	0.1329	0.1474	0.1608	0.1728	0.1833	0.1927	0.2009	0.2082	0.2147
	GRU-LSTM Autoencoder	0.0631	0.0841	0.1058	0.1261	0.1431	0.1574	0.1693	0.1794	0.1882	0.1958	0.2124	0.2283
	LSTM-GRU	0.1087	0.1249	0.1429	0.1602	0.1768	0.1946	0.2115	0.2272	0.2397	0.2504	0.2607	0.2693
	LSTM-GRU Attention	0.0919	0.0998	0.11	0.1205	0.1323	0.1447	0.156	0.1686	0.179	0.1886	0.1969	0.205
	Seq2Seq GRU Attention (Ours)	**0.0536**	**0.0705**	**0.0885**	**0.1052**	**0.1207**	**0.1348**	**0.1474**	**0.1586**	**0.1684**	**0.1773**	**0.1853**	**0.1925**

The bold-font numbers represent the data in this study.

**Table 9 sensors-24-05069-t009:** Comparative analysis of average forecasting errors (1st to 12th time steps) of various models for all the primary pollutants (PM_2.5_, PM_10_, NO_2_, SO_2_, CO, and O_3_).

Models	Metrics	PM_2.5_	PM_10_	NO_2_	SO_2_	CO	O_3_	Total Errors of All the Pollutants	Average Errors of All the Pollutants
GRU	RMSE	11.136	15.9095	14.5642	2.1399	0.3026	11.2254	55.2776	9.2129
	MAE	8.6381	12.4437	11.0778	1.6396	0.2435	8.6886	42.7313	7.1219
	MAPE	65.927	63.6499	36.9947	21.8729	39.5353	39.8009	267.7807	44.63
	R2	−0.0202	0.2539	0.4458	−0.0327	−0.8734	0.7126	0.486	0.081
	Theil’s U1	0.2312	0.2429	0.1943	0.1762	0.3231	0.1929	1.3606	0.2268
LSTM-GRU	RMSE	9.2383	12.1557	15.2289	2.0349	0.3114	10.2841	49.2533	8.2089
	MAE	7.1525	9.2615	11.722	1.5903	0.2589	7.9149	37.9001	6.3167
	MAPE	44.0939	43.442	32.2471	20.5464	42.941	40.2095	223.4799	37.247
	R2	0.3159	0.5651	0.3959	0.0653	−0.9833	0.7581	1.117	0.1862
	Theil’s U1	0.1992	0.1972	0.207	0.1702	0.355	0.1734	1.302	0.217
Seq2Seq GRU	RMSE	8.3895	12.4145	14.0746	2.2411	0.1988	10.3611	47.6796	7.9466
	MAE	6.3549	8.5585	11.1305	1.7289	0.1369	7.7941	35.7038	5.9506
	MAPE	32.0781	42.8591	32.3989	22.8036	22.1323	54.698	206.97	34.495
	R2	0.4239	0.5477	0.35	−0.158	0.183	0.7022	2.0488	0.3415
	Theil’s U1	0.1853	0.2012	0.2185	0.1859	0.2073	0.1938	1.1920	0.1987
GRU Autoencoder	RMSE	7.4172	9.9001	15.6408	2.1783	0.2087	10.072	45.4171	7.5695
	MAE	5.5559	7.338	11.996	1.7238	0.1505	7.723	34.4872	5.7479
	MAPE	26.6251	32.1267	31.6768	22.886	25.6518	45.5455	184.5119	30.752
	R2	0.5576	0.7142	0.3584	−0.0928	0.098	0.769	2.4044	0.4007
	Theil’s U1	0.1635	0.1638	0.2198	0.1803	0.2106	0.1677	1.1057	0.1843
GRU-LSTM Autoencoder	RMSE	6.795	10.8646	13.3295	1.8162	0.2068	8.7269	41.739	6.9565
	MAE	4.9944	8.6353	10.3939	1.4307	0.1397	6.5688	32.1628	5.3605
	MAPE	24.3318	39.2718	29.4902	19.25	23.612	35.7451	171.7009	28.617
	R2	0.622	0.6535	0.5282	0.2352	0.1032	0.8205	2.9626	0.4938
	Theil’s U1	0.154417	0.1748	0.1832	0.1511	0.2161	0.1462	1.025817	0.171
GRU Attention	RMSE	8.3713	11.0648	14.4883	2.253	0.2273	9.3958	45.8005	7.6334
	MAE	6.2751	8.2416	11.4436	1.8091	0.1781	6.8841	34.8316	5.8053
	MAPE	37.7368	35.0587	38.1274	25.2845	28.994	37.7002	202.9016	33.817
	R2	0.4248	0.6346	0.4481	−0.1672	−0.0578	0.7984	2.0809	0.3468
	Theil’s U1	0.1815	0.1792	0.2	0.1821	0.2433	0.156	1.1421	0.1904
LSTM-GRU Attention	RMSE	7.9259	8.9586	14.2653	1.756	0.2696	9.7584	42.9338	7.1556
	MAE	5.9051	6.5493	10.7119	1.3087	0.2169	7.3914	32.0833	5.3472
	MAPE	31.577	27.9649	36.675	16.7207	34.8153	39.7733	187.5262	31.254
	R2	0.4961	0.7657	0.4717	0.3001	−0.4872	0.7842	2.3306	0.3884
	Theil’s U1	0.1742	0.1494	0.1897	0.1503	0.3026	0.1618	1.128	0.188
Seq2Seq LSTM Attention	RMSE	7.8098	9.5764	14.9496	1.9911	0.2077	10.6169	45.1515	7.5252
	MAE	5.9441	7.1611	11.6294	1.5863	0.1466	8.2088	34.6763	5.7793
	MAPE	35.6981	32.3812	33.6035	21.1985	25.1343	38.8983	186.9139	31.1523
	R2	0.5181	0.7327	0.4134	0.0977	0.1151	0.7359	2.6129	0.4354
	Theil’s U1	0.1738	0.1599	0.2071	0.1653	0.2112	0.1848	1.1021	0.1836
Seq2Seq Bi-LSTM Attention	RMSE	6.921	9.101	13.6094	1.7209	0.1843	9.0814	40.818	6.803
	MAE	5.2668	6.5749	10.5807	1.3537	0.128	7.8004	31.7045	5.2841
	MAPE	30.005	28.0484	34.9105	17.8384	21.3053	41.2917	173.3993	28.8998
	R2	0.5951	0.7482	0.5112	0.3136	0.2934	0.7678	3.2293	0.5382
	Theil’s U1	0.1606	0.153	0.1834	0.1451	0.1901	0.1679	1.0001	0.1667
Seq2Seq GRU Attention (Ours)	RMSE	**5.5078**	**8.0605**	**11.1184**	**1.3741**	**0.1590**	**7.1258**	**33.3458**	**5.5576**
	MAE	**3.1959**	**4.7071**	**7.5100**	**0.9111**	**0.0935**	**4.5675**	**20.9854**	**3.4975**
	MAPE	**18.2412**	**18.7405**	**23.4140**	**12.6067**	**16.9963**	**25.1959**	**115.1948**	**19.1991**
	R2	**0.7523**	**0.8021**	**0.6733**	**0.5686**	**0.4776**	**0.8815**	**4.1557**	**0.6926**
	Theil’s U1	**0.1253**	**0.1335**	**0.149**	**0.1183**	**0.1633**	**0.1214**	**0.8109**	**0.1351**

The bold-font numbers represent the data in this study.

**Table 10 sensors-24-05069-t010:** TensorFlow and TensorFlow Lite models file size comparison.

Properties	Original TF Model	TFLite Model (without Quantization)	TFLite Model— Dynamic Range Quantization	TFLite Model— Full-Integer Quantization (Integer with Float Fallback)	TFLite Model—Full-Integer Quantization (Integer-Only)	TFLite Model— Float16 Quantization
File size (KB)	1176	397	119	125	125	206

**Table 11 sensors-24-05069-t011:** File size comparison TF and TFLite models.

Properties	Original TF Model	TFLite Model (without Quantization)	TFLite Model— Dynamic Range Quantization	TFLite Model— Full-Integer Quantization (Integer with Float Fallback)	TFLite Model—Full-Integer Quantization (Integer-Only)	TFLite Model— Float16 Quantization
File size (KB)	1176	397	119	125	125	206

**Table 12 sensors-24-05069-t012:** Comparison of execution time (seconds) for TFLite models.

Properties	Original TF Model	TFLite Model (without Quantization)	TFLite Model— Dynamic Range Quantization	TFLite Model— Full-Integer Quantization (Integer with Float Fallback)	TFLite Model—Full-Integer Quantization (Integer-Only)	TFLite Model— Float16 Quantization
Execution time (seconds)	323.597737	52.671870	59.980179	68.883801	65.837953	54.748388

**Table 13 sensors-24-05069-t013:** Comparison of average model accuracy of the six pollutants across 12 time steps for the original TF models and TFLite models.

Metrics	Original TF Model (without Quantization)	TFLite Model (without Quantization)	TFLite Model— Dynamic Range Quantization	TFLite Model— Full-Integer Quantization (Integer with Float Fallback)	TFLite Model— Full-Integer Quantization (Integer-Only)	TFLite Model— Float16 Quantization
RMSE	5.5576	7.9016	8.0783	9.9083	8.8866	8.1815
MAE	3.4975	4.385	4.5146	5.1466	4.8433	4.5883
MAPE	19.1991	22.0133	23.755	25.8483	20.8483	24.09
R2	0.6926	0.6566	0.6583	0.5283	0.5466	0.635
U1	0.135	0.1443	0.1466	0.155	0.1516	0.1483

## Data Availability

Data are available in a publicly accessible repository at “https://github.com/DivyaBharathi18/Fog-enabled-Air-Quality-Monitoring-and-Prediction”, accessed on 25 July 2024. The air quality dataset is derived from the resources available in the CPCB (Central Pollution Control Board, India) website: https://airquality.cpcb.gov.in/ccr/#/caaqm-dashboard-all/caaqm-landing/caaqm-data-repository, accessed on 25 July 2024.

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
