# Peer review of "Design and Enhancement of a Fog-Enabled Air Quality Monitoring and Prediction System: An Optimized Lightweight Deep Learning Model for a Smart Fog Environmental Gateway"

_sensors, 2024, doi:10.3390/s24155069_

Round 1
Reviewer 1 Report
Comments and Suggestions for Authors
See attached file.

Reviewer 2 Report
Comments and Suggestions for Authors
This paper presents an innovative approach to air quality monitoring and forecasting, leveraging IoT, Fog Computing, and Deep Learning. The authors have developed a Fog-enabled Air Quality Monitoring and Prediction (FAQMP) system that aims to provide real-time, accurate air quality data and forecasts in smart city environments.
The system's well-thought-out architecture comprises three layers:
1. A sensing layer with low-cost sensors
2. A fog computing layer featuring a Smart Fog Environmental Gateway (SFEG)
3. A cloud computing layer for long-term storage and complex analysis
Key strengths:
· Cost-effectiveness: The use of low-cost sensors and optimized data transmission makes large-scale deployment feasible in places like India.
· Real-time processing: Fog computing brings computation closer to the data source, enabling faster processing and response times.
· Efficient communication: The integration of LoRa technology for long-range, low-power communication is well-suited for IoT applications.
The core of the system is its Deep Learning model - a Seq2Seq GRU Attention model for multivariate multi-step air quality forecasting. The authors have done a commendable job in developing and optimizing this model for deployment on resource-constrained fog nodes. The comparative analysis with baseline models is thorough and convincing, showing clear improvements in forecasting accuracy.
Constructive Criticisms and Suggestions:
1. Model Complexity vs. Resource Constraints: While the optimized lightweight model is impressive, a comparison with simpler models (e.g., machine learning or statistical models) in terms of performance vs. resource usage would be valuable, especially for extremely resource-constrained environments.
2. Scalability and Real-world Deployment: The paper would benefit from more discussion on the practical challenges of scaling this system to a city-wide level. What are the potential bottlenecks? How would the system handle thousands of sensors?
3. Data Quality and Sensor Calibration: More details on ensuring long-term data quality would be valuable, particularly addressing sensor drift in high-temperature and dusty environments typical in India.
4. Long-term Performance: While the paper presents impressive results, more discussion on how the system handles seasonal changes or long-term trends in air quality would be beneficial.
5. Interdisciplinary Impact: Expanding on the potential impact of this system on public health policies or urban planning decisions would enhance the paper's broader relevance.
General Comments:
6. Conciseness: The paper is well-written but quite long. Consider condensing some sections, particularly generic definitions, to improve readability. For instance, the background on IoT and Fog Computing (pages 3-4) could focus more on their specific applications to air quality monitoring.
7. Figures: While informative, some figures (like Fig. 10 and 11) are dense and difficult to read. Consider splitting these into multiple figures for clarity. For example, Figure 10 could be divided into separate figures for error metrics (RMSE and MAE) and performance metrics (R2 and Theil's U1).
8. Real-world Impact: More discussion on potential real-world impact and implementation challenges would enhance the paper's practical relevance.
Detailed Suggestions:
9. Abstract: The abstract, while informative, could be more concise. For example: "This enables real-time processing, accurate forecasting, and timely warnings of dangerous AQI levels while optimizing fog resources."
10. Introduction: Improve paragraph transitions. For instance, between paragraphs 2 and 3 on page 2, consider adding: "To address these challenges, researchers have turned to advanced technologies such as IoT and Fog Computing."
11. Literature Review: Make the review more critical. When discussing existing AQM systems (pages 5-6), analyze their limitations. For example: "While Kumar et al. [25] developed a cloud-based AQM system, it lacks real-time processing capabilities crucial for immediate response to air quality changes."
12. Methodology: The explanation of the Seq2Seq GRU Attention model (pages 21-23) could benefit from a simplified analogy for non-experts. For instance: "Think of the encoder as a reader summarizing a book, the attention mechanism as highlighting important passages, and the decoder as an author writing a new chapter based on those highlights."
Despite these suggestions, the depth of this work is impressive. The authors have clearly put significant thought into every aspect of the system, from hardware choices to software optimizations. The EnviroWeb application is a commendable addition, making the data accessible and understandable to end-users. The optimization of the Deep Learning model for fog deployment is particularly noteworthy.
This paper makes a significant contribution to the field, bridging the gap between theoretical models and practical implementation in smart city environmental monitoring. With revisions addressing the points raised, this paper could be a highly impactful publication in the field.
Comments on the Quality of English LanguageStructural issues, should be easy to address as mentioned above.
Round 2
Reviewer 1 Report
Comments and Suggestions for Authors
The paper notably improved after its revision, and I have no further comments.